# Modeling enculturated bias in entrainment to rhythmic patterns

**Thomas Kaplan**[1¤]*, **Jonathan Cannon**[2], **Lorenzo Jamone**[1,3], **Marcus Pearce**[1,4]

**1** Cognitive Science Research Group, School of Electronic Engineering & Computer Science, Queen Mary University of London, London, United Kingdom, **2** Department of Psychology, Neuroscience & Behaviour, McMaster University, Hamilton, Ontario, Canada, **3** Advanced Robotics at Queen Mary (ARQ), School of Electronic Engineering & Computer Science, Queen Mary University of London, London, United Kingdom, **4** Department of Clinical Medicine, Aarhus University, Aarhus, Denmark

¤ Current address: School of Electronic Engineering and Computer Science, Queen Mary University of London, London, United Kingdom, E1 4NS
* t.m.kaplan@qmul.ac.uk

**Data Availability Statement:** All data and code used for running experiments, analysis and plotting are available on an OSF repository (https://doi.org/10.17605/osf.io/tpwfn) and in Supporting information files.

## Abstract

Long-term and culture-specific experience of music shapes rhythm perception, leading to enculturated expectations that make certain rhythms easier to track and more conducive to synchronized movement. However, the influence of enculturated bias on the moment-to-moment dynamics of rhythm tracking is not well understood. Recent modeling work has formulated entrainment to rhythms as a formal inference problem, where phase is continuously estimated based on precise event times and their correspondence to timing expectations: PIPPET (Phase Inference from Point Process Event Timing). Here we propose that the problem of optimally tracking a rhythm also requires an ongoing process of inferring which pattern of event timing expectations is most suitable to predict a stimulus rhythm. We formalize this insight as an extension of PIPPET called pPIPPET (PIPPET with pattern inference). The variational solution to this problem introduces terms representing the likelihood that a stimulus is based on a particular member of a set of event timing patterns, which we initialize according to culturally-learned prior expectations of a listener. We evaluate pPIPPET in three experiments. First, we demonstrate that pPIPPET can qualitatively reproduce enculturated bias observed in human tapping data for simple two-interval rhythms. Second, we simulate categorization of a continuous three-interval rhythm space by Western-trained musicians through derivation of a comprehensive set of priors for pPIPPET from metrical patterns in a sample of Western rhythms. Third, we simulate iterated reproduction of three-interval rhythms, and show that models configured with notated rhythms from different cultures exhibit both universal and enculturated biases as observed experimentally in listeners from those cultures. These results suggest the influence of enculturated timing expectations on human perceptual and motor entrainment can be understood as approximating optimal inference about the rhythmic stimulus, with respect to prototypical patterns in an empirical sample of rhythms that represent the music-cultural environment of the listener.

**Funding:** TK was supported by a doctoral studentship from the Engineering and Physical Sciences Research Council (EPSRC, https://epsrc.ukri.org/) and Arts and Humanities Research Council (AHRC, https://ahrc.ukri.org/) Centre for Doctoral Training in Media and Arts Technology (EP/L01632X/1). The funders had no role in study design, data collection and analysis, decision to publish, or preparation of the manuscript.

**Competing interests:** The authors have declared that no competing interests exist.

## Author summary

Cross-cultural studies have highlighted that listeners from non-Western cultures can precisely tap along with complex rhythms present in music from their culture that are challenging for participants from Western cultures. Therefore, while most adults can synchronize movements with simple periodic patterns (e.g. a ticking clock, a metronome), the ability to precisely track more complex rhythmic patterns depends on musical experience. Many computer models have been developed to describe the remarkable precision of human "entrainment", but they have done little to explain how this ability depends on cultural musical experience. Here, we describe this as the problem of estimating the phase of a cycle underlying an auditory rhythm in real time, by drawing upon learned patterns (reference structures) that could plausibly describe the structure of observed events. By creating a model that solves this inference problem, and configuring these patterns to reflect specific musical features, we are able to simulate cultural variation in synchronization to rhythm. These results highlight that while humans universally move to musical rhythm, the ability to do so depends on musical experience within a cultural tradition, as reflected by the distinct "categories" of rhythm learned during such experience.

## Introduction

Humans are remarkably skilled at detecting temporal patterns in auditory stimuli such as speech and music. Expressive performances of a musical score can deviate greatly in the precise timing of musical events, yet listeners are still able to identify prototypical rhythms, which can be notated [1] and associated with synchronized movement [2]. The precision of this synchronization varies based on a listener's musical experience: long-term music-cultural exposure shapes rhythm perception, leading to enculturated expectations that make certain (culturally-familiar) rhythms easier to predict and more conducive to synchronized movement. This has been observed in finger-tapping tasks that highlight the stability with which participants from India [3], Turkey [4] and Mali [5] synchronize to complex rhythms present in their cultures, in contrast to participants from Western cultures in which these rhythms are uncommon [6, 7]. The present paper focuses on the cognitive processes underlying this enculturated bias in *entrainment*. Entrainment typically refers to the physical principles of mode-locking between coupled oscillating systems [8], but here we define entrainment empirically as in [9]: the temporal alignment of a biological or behavioral process with the regularities in an exogenously occurring stimulus. In humans, this includes observed synchronization of movement to predictable rhythmic patterns such as isochronous or non-isochronous beats found in different musical cultures [10, 11]. In order to model this behavior, we adopt a recent theoretical perspective which describes entrainment computationally as a dynamic inference of stimulus phase based on an internal model of rhythmic structure [9].

There is a history of using probabilistic models to simulate perception and production of rhythmic phenomena [2, 9, 12–16], and to describe rhythmic structure (e.g. [17, 18]). In particular, the effect of musical enculturation on listeners' expectations has been modeled as coupled cognitive processes of statistical learning and probabilistic prediction [16], whereby listeners learn discrete grammars characterizing the probabilistic structure of experienced musical styles (as proposed in [19]). One application includes the perception of *metrical structure* [14, 15], a hierarchically-embedded collection of recurring temporal periods inferred from and aligned to a stimulus rhythm [20]. These probabilistic models are reliant on symbolic sequence-learning algorithms [21], learning from (symbolic) score-based musical corpora.

While there is a long tradition of cross-cultural music corpus studies [22, 23], these models are limited by a coarse representation of rhythm, and cannot explain how people dynamically map—during entrainment—between auditory rhythms represented in a continuous space and discretized (symbolic) representations. As these models cannot serve as models of real-time behavior, there is considerable room for improvement in our computational understanding of how probabilistic temporal expectations continuously bias entrainment.

Recent psychological experiments have yielded data that are especially challenging for existing probabilistic models of rhythm perception. Jacoby et al. [24, 25] investigated discrete perceptual representations of auditory rhythm through serial reproduction tasks, providing substantial evidence that the perception of auditory rhythm is biased towards distinct rhythmic categories [26–28]; and that these categories vary cross-culturally. In these experiments, participants were asked to synchronize tapping with cyclical rhythms composed of three random time intervals, and the synchronized responses were averaged to produce a three-interval stimulus rhythm for the next iteration. After several iterations these rhythms were biased towards distinct integer ratios between successive intervals (see Fig 1). For all participant groups tested across five continents and fifteen countries [25] these categories included small integer-ratio rhythms, such as an isochronous pattern (intervals of ratios 1:1:1) and long-short interval patterns (cyclic rotations of ratios 2:1:1). This widespread preference for small integer-ratio rhythms appears to reflect universal biases in human auditory perception [29–32]. There was also substantial variation in the perceptual categories (rhythmic patterns) measured, which was clearly related to the prior music-cultural experience of participants. Traditional musicians demonstrated significantly greater bias towards mildly complex and culturally relevant rhythmic patterns when compared to students, for example in the Malian (ratios 3:3:2) and Turkish (ratios 2:2:3) groups. Crucially, this suggests that the discrete representations of rhythm that individuals draw upon during entrainment are contingent on their enculturation, and not fixed by biological constraints.

While the tendency to perceive and (re)produce continuously timed rhythms categorically in terms of simple integer ratio intervals is well documented [1, 24–27], the reasons for it are not well understood. This tendency might reflect intrinsic physiological dynamics. For example, these simple ratios offer stability for coupled neural oscillators given intrinsic mode-locking properties [33, 34]. Cross-frequency coupled oscillator models have been used to simulate

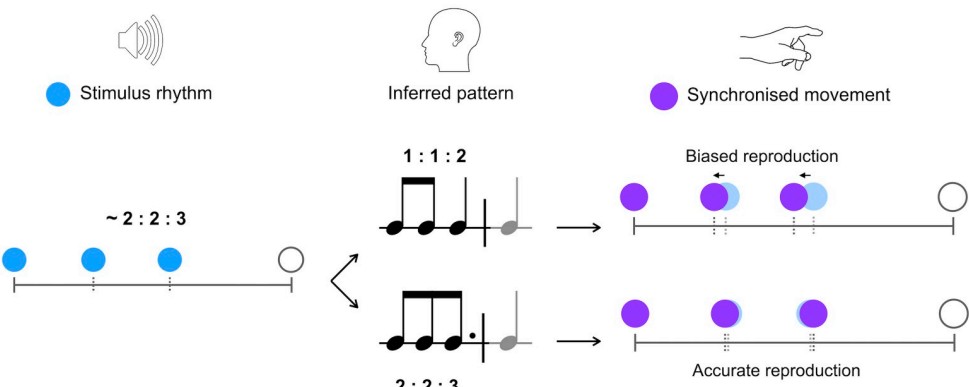

**Fig 1. Perceptual bias in entrainment.** Entrainment to an auditory rhythm through synchronized movement reveals a biased response, as in Bayesian inference, towards a listener's perceptual priors [24, 25]. If a listener does not have enculturated familiarity with musical rhythms related by the ratios 2:2:3, they are likely to interpret a rhythm (approximately) related by these ratios as 1:1:2, and this will be reflected in synchronized movements that are in fact closer to ratios 1:1:2 than 2:2:3.

this preference in iterated reproduction [35] and rhythm categorization [36], but neither offered an explanation for enculturated biases towards specific ratios. However, oscillator models with Hebbian plasticity [37] are capable of learning complex rhythmic patterns through frequent exposure [38, 39]. This suggests enculturation effects in entrainment might be explained using such (low-level) neural mechanisms, but it has not been shown that these results generalize to the cultural differences observed in iterated reproduction studies [24, 25].

The purpose of the present research is instead to investigate whether it is possible to simulate listeners' expectations, including both cross-culturally universal tendencies towards simple integer ratios and culturally-acquired deviations towards specific (mildly) complex integer ratio rhythms, solely through learning the statistics of interval patterns in music of these cultures (i.e., without any pre-existing bias for simple-integer ratios). For that reason we focus on the distinction between simple and complex integer ratio rhythms, rather than integer and non-integer ratio rhythms. Hence, in the present work, listeners' expectations are modeled using real-world notated musical rhythms, which consist solely of integer interval ratios. In the Discussion we consider the plausibility of this approach, with respect to intrinsic biological constraints of motor processes and temporal perception.

In order to define this perceptual problem of enculturated biases towards specific integer interval ratios in entrainment at a satisfactory computational level, we start by adopting a formal definition of temporal expectation and entrainment, namely "Phase Inference from Point Process Event Timing" (PIPPET) [9]. PIPPET describes expectation-based entrainment as an inference problem which affords a precise solution: inferring the state of an exogenous process that generates a series of events in time. This involves dynamically estimating phase and uncertainty (inverse of precision), where phase is a hidden variable denoting progress through an expected sequence of events. PIPPET is consistent with the "predictive-processing" framework, which proposes that the brain continuously attempts to infer the hidden causes of sensory events using a learned understanding (*generative model*) of how those causes produce sensations [40]. Estimates of hidden causes are optimized by minimizing the *prediction error* between new sensory information and predictions based on current estimates, which is achieved by Bayesian inference on the generative model parameters. Prediction errors are weighted by the precision (certainty) of predictions, to moderate new observations against prior expectations about the hidden causes, such that greater prediction errors would be assigned to prior expectations with high certainty (precision). The influence of prior expectations in this Bayesian formulation of entrainment exerts an inferential bias on timing predictions resembling the perceptual bias observed in iterated reproduction paradigms. For the purposes of this work, PIPPET also grants flexibility in the definition of temporal expectations acquired by enculturated listeners.

However, one critical limitation of PIPPET is that entrainment is performed in the context of just one expected rhythmic pattern, so it cannot describe how a listener matches a pattern of expectations with an ambiguous rhythmic stimulus. Depending on the temporal organization of a stimulus, it is clear from iterated reproduction studies that listeners experience perceptual biases towards multiple rhythmic patterns (categories). Listeners must therefore infer a suitable interpretation of an ambiguous stimulus rhythm, which involves drawing upon prior listening experience (as in Fig 1). One example of this is in entrainment to polyrhythms, overlapping combinations of rhythmic patterns, where listeners identify just one of the rhythmic streams as a perceptual "ground" [20, p. 48] when they are unable to integrate the parallel rhythmic streams. Even expert musicians struggle to track multiple cyclically structured sound streams [41, 42], suggesting that listeners both dynamically assess the possible interpretations of an auditory stimulus, and then draw upon the most plausible to form a single estimate of phase. However, this assessment must be updated dynamically during

perceptual entrainment—otherwise, we would not have the flexibility to switch to a different interpretation as a rhythm changes [3]. How do listeners dynamically assess the plausibility of different rhythmic patterns underlying some auditory stimulus, while at the same time drawing on their expectations to inform and bias their entrainment?

In the present paper, we address this question by incorporating rhythmic pattern selection for expectancy-based entrainment in an extension of PIPPET, which we call **p**PIPPET (PIP-PET with **p**attern inference). Accordingly, we present variational filtering equations that approximate a perfect Bayesian solution to this problem. Compared with existing entrainment models, two new elements are introduced: (1) dynamic estimation of which discrete rhythmic pattern provides the best interpretation of a sequence of continuous-time auditory events; and (2) dynamic weighing of these interpretations to inform a single estimate of phase.

In the Results section we use pPIPPET to simulate the results of existing empirical datasets. In three experiments we examine whether the observed performance of participants can be related to specific differences in prior expectations, and their influence on expectancy-based entrainment, as characterized by pPIPPET. In Experiment 1 we use the model to simulate listeners from distinct musical cultures synchronizing to two-interval rhythms [5], i.e. repeating cycles of two sounds, defined by two inter-onset time intervals. In Experiment 2 we extend the scope of the simulations to periodic three-interval rhythms in a large rhythm space, which were explicitly notated (using discrete interval categories) by Western musicians [1]. In Experiment 3 we simulate enculturated bias in iterated reproduction of rhythms by listeners from several musical cultures, using three-interval rhythms from the same rhythm space as Experiment 2 [24, 25]. By deriving prior expectations in these simulations from musical corpora, we demonstrate that pPIPPET can successfully simulate enculturation effects on rhythm perception and production, integrating cultural priors, production constraints and universal perceptual biases. In the Discussion section we highlight the contributions of pPIPPET to cross-cultural research in musical rhythm perception and expand on the proposed implementation of PIPPET in the brain.

## Overview of pPIPPET

We briefly describe pPIPPET in this section. Please refer to the Methods section for detailed descriptions of both PIPPET and pPIPPET, and see S2 Text for the respective derivations.

pPIPPET directly extends PIPPET [9], the formal problem of dynamically estimating the phase underlying a sequence of timed events generated as a point process. As in [9] we solve this problem through continuous variational inference: a generative model is created to describe the probabilistic generation of events, and variational Bayes is used to continuously estimate phase and phase uncertainty of this generative model. Whereas PIPPET's generative model describes expectations for a single series of events, i.e. one rhythmic pattern, pPIPPET generates expectations for multiple rhythmic patterns.

Note that PIPPET is formulated as a perceptual process, and by extension pPIPPET is as well. While we do not specify precisely how entrained movement is produced by this process, we posit that perceptual and motor entrainment depend on the same internal process of estimating underlying phase (see [9, p. 21]). In other words, we model how the listener would interpret the rhythm in light of their expectations, which we assume would inform their reproduction of the rhythm.

**PIPPET.**    A listener's event-based expectations in PIPPET are modeled by positing that they expect events to occur as an inhomogenous point process, with probability $\lambda(\varphi)$, referred to as an *expectation template*. This consists of a summation of Gaussian distributions, each centered around a mean phase at which an event is expected, and scaled according to

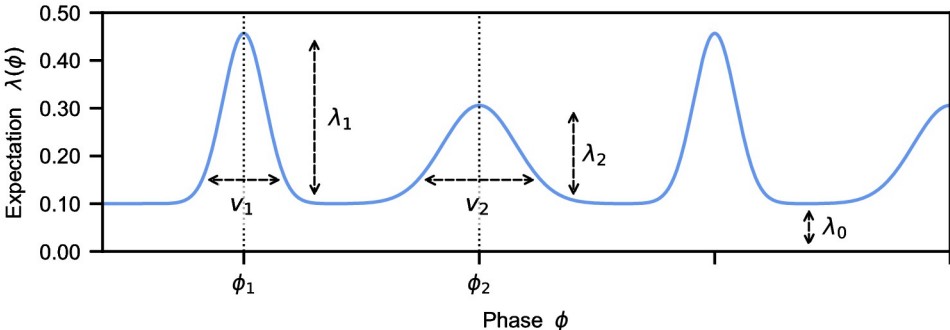

**Fig 2. PIPPET expectation template.** An expectation template in PIPPET, as defined in Eq (1), represents the instantaneous rate of events $\lambda(\phi)$ at phase $\phi$ of the underlying process. Expectations for specific events are each represented by a Gaussian centered at the respective phase $\phi_i$, with variance $v_i$ representing the precision of the expectation, and scaling factor $\lambda_i$ representing the strength of expectations. The constant $\lambda_0$ accounts for the rate of events unrelated to phase, i.e. background noise.

expectation strengths and precisions. This allows expectations for a precisely timed rhythmic pattern to be represented continuously, as in Fig 2.

The optimal Bayesian solution to rhythm tracking is approximated by a filter that dynamically estimates phase. The filter uses the expectation template to inform its estimate of phase based on the presence or absence of events in each $dt$ time step, pushing it toward expected event phases at each event. It maintains a Gaussian posterior for phase characterized by a mean $\mu_t$ and variance $V_t$ at any time $t$. Between events, the mean increases at a steady rate (though can deviate slightly from this trajectory when expectations are especially strong, see [9, Fig 7]), and the variance accumulates. At events, the posterior is updated using a precision-weighted sum of the phase as estimated from the stimulus history, and the phase at which an event is expected to occur. This allows PIPPET to make appropriate phase shifts on specific events according to prior expectations, and these shifts can be related to specific production biases in finger-tapping experiments (e.g. [5, 24, 25])—this would not be possible if assuming that phase advances steadily.

Note that while PIPPET can entrain to a periodic pattern, it does *not* detect periodicity as defined by the time interval between beats (tempo). Joint phase and tempo inference is addressed in PATIPPET, a variant of PIPPET which additionally infers the rate of phase advance over time [9]. In this work we assume a constant tempo, and consider how tempo might be incorporated in the Discussion.

Further to the original formulation of PIPPET, we have introduced two types of noise: (1) timing of stimulus events are perturbed to represent noisy delays in auditory processing; and (2) the estimated phase mean is perturbed to reflect the noisiness of timekeeping processes in the brain. Please refer to the Methods for details.

**pPIPPET.** In pPIPPET, a listener's event-based expectations are modeled by positing that they expect one of a set of rhythmic patterns (i.e. expectation templates), indexed by $m$, to be chosen at random with probabilities $p^m$, and for events to then occur as an inhomogenous point process based on the respective expectation template, $\lambda^m(\phi)$. The listener draws upon all templates to dynamically estimate the phase underlying observed events. A single estimate is maintained as a Gaussian posterior (as in PIPPET), instead of separate estimates per expectation template, given that listeners appear to track just one cyclically structured pattern in an ambiguous sound stream (described in the Introduction with respect to polyrhythms, also see [20, p. 48]).

Each template starts with a prior probability $p_0^m$ of generating an observed rhythm. At any time $t$, the posterior probability of each template $p^m$ is revised based on the most recent observation (presence or absence of an event in the current $dt$ time step), according to how much an event is anticipated by that template. The current posterior distribution for phase is then used to calculate posterior distributions in the context of each template $m$ (as in PIPPET), and these estimates are marginalized to approximate a single Gaussian posterior for phase. This is described in more detail in the Methods section with detailed derivations given in S2 Text.

In the following section we configure the pPIPPET filter using different sets of expectation templates, to investigate the resulting precision of entrainment. We relate this to differences in human entrainment as a function of musical culture, and how this reflects differing discrete expectations for rhythmic patterns between listeners. We do this by qualitatively comparing the results of our simulations to those of empirical studies [1, 5, 25] as the data is not publicly available.

Note that the results presented do not essentially depend on the configuration schemes used. The choices made should be viewed as starting points in applying pPIPPET to guide the interpretation of experimental data.

Parameters for these simulations can be found in S1 Text. Supplementary materials including the simulation code are available at doi.org/10.17605/osf.io/tpwfn.

## Results

### Experiment 1: Synchronization using culturally relevant rhythmic prototypes

We start by illustrating the basic behavior of the pPIPPET filter, simulating entrainment to simple periodic two-interval rhythms (i.e., a rhythm consisting of three events separated by two temporal intervals). Repeated two-interval rhythms can be considered basic elements of more complex rhythmic patterns. When presented with two-interval rhythms with unbalanced duration ratios (e.g. 3:1 or 3:2), synchronized tapping in participants reveals a perceptual bias, in that the tapped rhythm is typically pulled towards an approximate 2:1 ratio (see [43] for a review). This classic literature, exclusively studying Western participants, suggested that rhythm perception is dominated by two discrete rhythmic patterns (categories): balanced (1:1) and unbalanced (2:1) rhythms [26]. Recently, Polak et al. [5] demonstrated that these *rhythmic prototypes* vary depending on musical-cultural background. When presented with two-interval rhythms of a 4:3 ratio and fast pattern period, a group of Malian percussionists were able to precisely synchronize tapping to this stimulus, whereas the synchronized responses of conservatory students from Germany and Bulgaria were systematically biased towards a 2:1 ratio. The authors proposed that this reflected the presence of a unique rhythmic prototype of ratio 4:3 for the Malian group, which characterizes the complex-ratio subdivisions of dance music from Mali [44].

Here we simulate this enculturated bias by configuring pPIPPET filters with and without a pattern of expectations for a 4:3 ratio rhythm. For consistency with Polak et al. [5], in place of a precise 4:3 ratio we use a complex 58:42 ratio (= 4.14:3), which was measured from non-isochronous subdivision timings in a relevant repertoire of Malian music [45]–though for simplicity we refer to this ratio as 4:3. Similarly to the method of Polak et al., pPIPPET filters were presented with two-interval rhythms of ratios 1:1, 4:3 and 2:1 (Fig 3A), each repeated twenty-five times. We configured two filters using different sets of expectation templates: the first with templates for 1:1 and 2:1 rhythms; and the second with an additional template for a 4:3 rhythm. In both models, all patterns of expectation were configured with an equiprobable prior $p_0^m$ for the simplicity of illustration, but we expect these priors would differ empirically

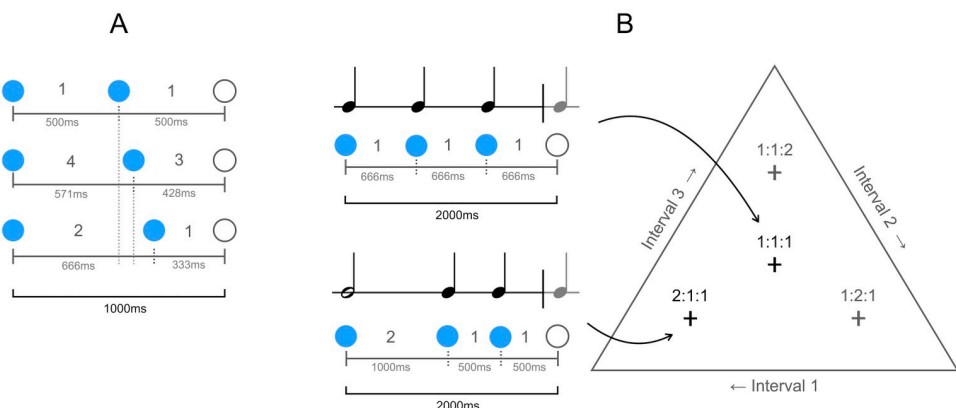

**Fig 3. Two- and three-interval rhythms.** A) Simple two-interval rhythms of ratios 1:1, 2:1 and 4:3, with a 1000ms period. B) Three-interval rhythm space, where each axis of the ternary plot refers to one of the three intervals in a rhythm. Each point corresponds to a three-interval rhythm of period 2000ms, with each interval constrained to a minimum duration of 300ms. The crosses denote some examples of rhythms related by small-integer ratios (e.g. 1:1:1 and 2:1:1, shown to the left).

due to learned factors and accordingly we derive them more systematically in the subsequent experiments. We refer to these filters as the (1) European and (2) Malian models, respectively. This reflects the participants simulated, who were recruited in either: (1) Germany and Buglaria, i.e. Europe; or (2) Mali. Note that this a convenient shorthand, and we are not stating that these models generalize to all individuals from Europe or Mali.

Fig 4 illustrates the tracking performance of these models for the first two repetitions of the 4:3 stimulus (see S2 and S3 Figs for the 1:1 and 2:1 stimuli). In absence of specific expectations for a 4:3 pattern (Fig 4A), the European model must draw upon other patterns of expectations (1:1 and 2:1) in order to track the stimulus. While the timing of stimulus events is closer to a 1:1 rhythm numerically, the stimulus events are attributed to the uneven 2:1 template, as the 2:1 template is configured with lower precision than the 1:1 template. The specificity (high precision) of the 1:1 expectations causes unevenly timed rhythms (e.g. 4:3) to be associated with the comparatively imprecise 2:1 expectations. We liken this effect to the widely observed tendency of listeners to associate uneven rhythms with a long-short pattern (i.e. 2:1, e.g. see [24, Fig S1] and [5, pp. 5-6]). This attractor effect causes the 2:1 template plausibility to increase rapidly over successive events, such that the 2:1 candidate posterior on phase is strongly weighted in the marginalized phase estimate. This causes the phase estimate to be corrected forwards and backwards when each event arrives early (first interval) or late (second interval) versus expectations, creating fluctuations in phase uncertainty. In contrast, the Malian model tracks phase successfully, as the expectations for a 4:3 pattern produce an accurate candidate posterior on phase (Fig 4B). However, it takes several repetitions for the probability of the 4:3 template to converge on maximum likelihood, causing the phase uncertainty to gradually decrease along with plausibility of the (inaccurate) 2:1 template. For all stimuli, the models' probability distribution over templates converge after five repetitions, and the ranking of template likelihoods converges within only two cycles (see S1–S3 Figs).

Phase corrections performed for the 4:3 rhythm are summarized in Fig 5A with respect to the first event in each cycle. It is clear that phase estimates in the European model are consistently skewed, with a distribution resembling the biased responses (synchronized tapping) of German and Bulgarian participants. The distributions of corrections for the Malian model demonstrate an almost unbiased response (i.e. precise synchronization), as shown by the Malian participants, suggesting the underlying stimulus pattern is correctly inferred.

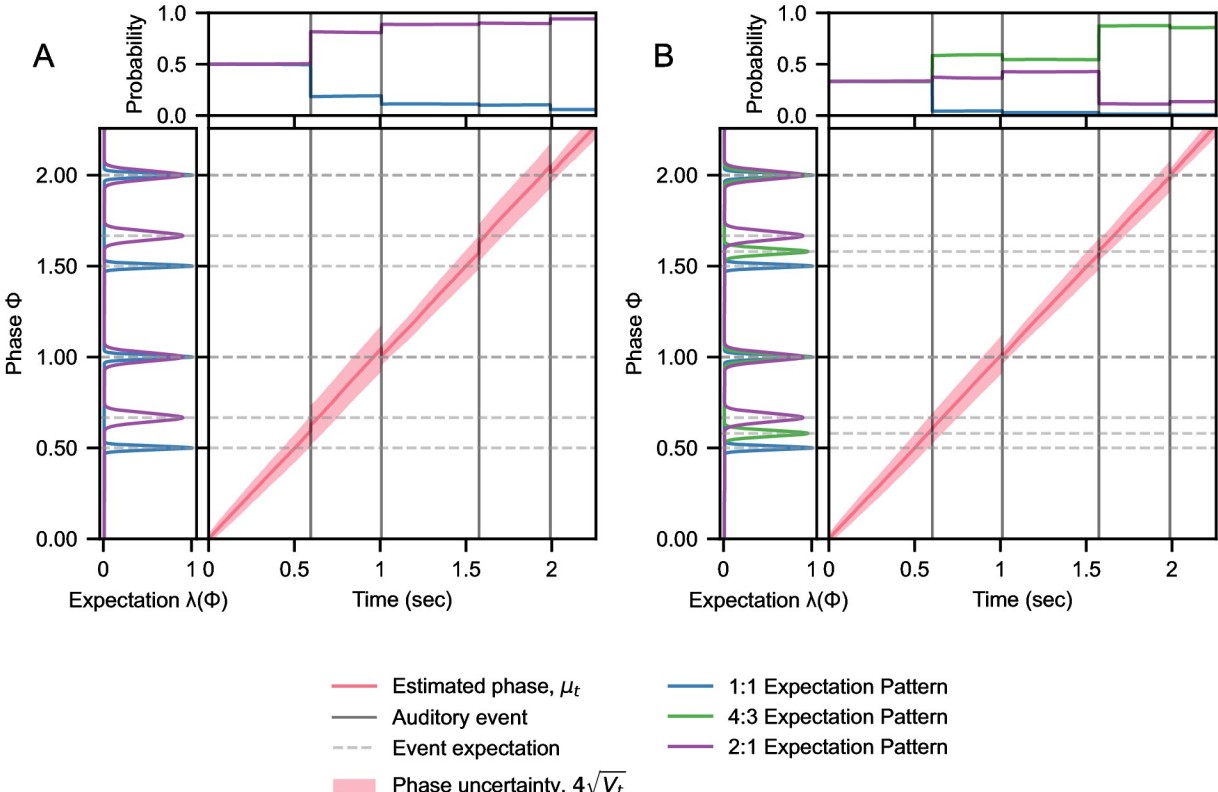

**Fig 4. Tracking the phase of a 4:3 rhythm with different timing expectations.** Two pPIPPET models are given patterns of expectations for 1:1 and 2:1 rhythms, but only one with expectations for 4:3 rhythms. The resulting quality of phase tracking—for the first two stimulus repetitions—is shown through adjustments to estimated phase $\mu_t$ on auditory events, alongside changes in uncertainty $V_t$. Implicit inference of the rhythmic pattern over time is shown through changes in template probability $p^m$. A) European model. Without 4:3 expectations, phase must be adjusted after the first event of each cycle to compensate for the timing shift, causing phase uncertainty to increase until the cycle is complete, when phase is shifted back. B) Malian model. Phase is successfully tracked, with phase uncertainty only growing slightly between events. Note that phase uncertainty always accumulates between events due to expected phase noise (see Methods, $\sigma$ in Eq 2).

Furthermore, the relative levels of phase uncertainty on time steps preceding auditory events (Fig 5B) are similar to average variation in tapping asynchronies for participants. Phase uncertainty for both models is lower for even (1:1) than uneven (4:3 and 2:1) rhythms, as per all participant groups. This was significant for both the Malian model (Welch's t-tests, 1:1 vs 4:3, $t(50) = -2.21$, $p = .030$; and 1:1 vs 2:1, $t(50) = -3.35$, $p = .001$) and European model (1:1 vs 4:3, $t(50) = -10.68$, $p < .001$; and 1:1 vs 2:1, $t(50) = -4.81$, $p < .001$). Additional comparisons of the uncertainty between uneven rhythms indicate greater consistency across stimuli for the Malian model ($t(50) = -1.58$, $p = 0.12$) than the European model ($t(50) = 5.48$, $p < .001$); resembling the consistently accurate tapping of the Malian participants. The difference between the two models for the 4:3 stimulus is substantial ($t(50) = -7.24$, $p < .001$), whereas the differences for the 2:1 stimulus were non-significant ($t(50) = .23$, $p = .82$), and weakly significant for the 1:1 stimulus ($t(50) = 2.28$, $p = .027$). If expectation variances and/or prior likelihoods were parameterized differently for the European and Malian models, such as higher relative precision of expectations for 2:1 and 1:1 templates in the European model, then (stronger) significant differences could emerge for the 2:1 and 1:1 stimuli; and these changes might be necessary to accurately model either Polak et al.'s German or Bulgarian participant groups, compared to our collective European model.

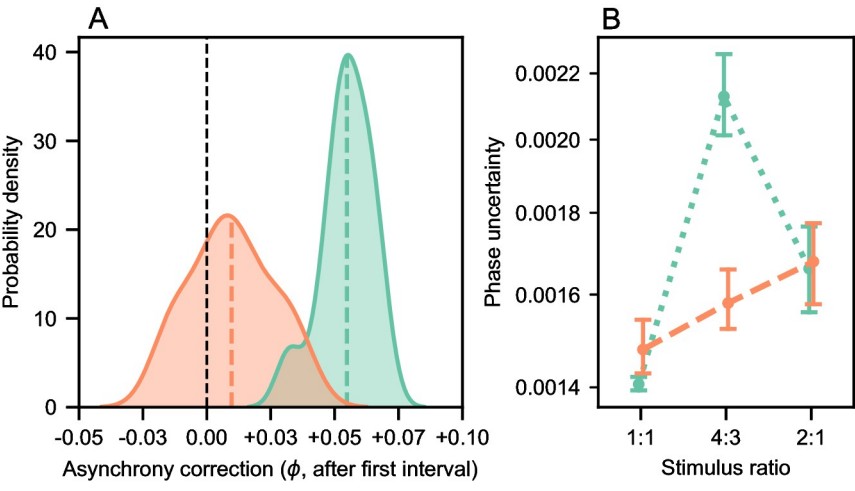

**Fig 5. Production bias when tracking 1:1, 4:3 and 2:1 rhythms.** Performance of pPIPPET in tracking repeating two-interval rhythms, depending on template configuration. A) Distribution of phase corrections following the first interval of the 4:3 stimulus rhythm. Curves are Gaussian kernel density estimates, vertical black dashed line shows an unbiased response, and other vertical dashed lines refer to sample means. B) Phase uncertainty on the time step preceding an auditory event. Error bars show the 95% confidence intervals within each sample. Note that this figure shows qualitative agreement with empirical tapping data shown in [5, Fig 3].

Together these results show that the systematic biases in entrainment observed by Polak et al. [5] can be explained in pPIPPET through configuration with different discrete sets of expectation templates. These expectation templates serve as explicit perceptual priors for rhythmic patterns and bias phase inference accordingly, such that pPIPPET provides a precise probabilistic interpretation of key high-level concepts like perceptual "prototypes" and "categories" for rhythm [26, 27] in the context of continuous entrainment.

## Experiment 2: Categorization of non-uniform rhythms

The previous experiment involved implicit inference of a temporal pattern underlying a rhythmic stimulus, but the breadth of the stimulus domain was limited. Now we expand the domain to a larger "rhythm space" consisting of three intervals (Fig 3B), and examine the mapping of this large continuous space of temporal patterns to the discrete space of expectation templates. In doing so, we simulate the perceptual categorization of non-uniform rhythms, i.e. rhythms not related by exact integer ratios. Following Desain & Honing [1], *categorization* here refers to the cognitive process of extracting discrete rhythmic categories from a continuous signal. Perceptual categories in this rhythm space were first uncovered by Desain & Honing [1] in an innovative task whereby musicians translated a grid of sampled three-interval rhythms into common music notation—a discrete, symbolic space. Across participants, the largest number of notated responses were for rhythms related by ratios of integers 1 and 2, i.e. 1:1:1 and 1:1:2 (and cyclic rotations), suggesting a perceptual attractor effect. Between participants, the responses were most consistent (lowest entropy) around these rhythms, with the attractor effects forming delineated quasi-convex shapes.

Here we apply a pPIPPET filter to this large rhythm space, measuring the relative levels of entropy in the discrete probability distribution over expectation templates. The rhythm space consists of three-interval patterns which each sum to a two second duration, with composing

intervals of at least 300ms (for consistency with Experiment 3), sampled at a resolution of 10ms ($n = 6216$). We demonstrate that the perceptual attractor effect observed as a static phenomenon in categorization tasks can be qualitatively reproduced by the dynamic inference process implemented in pPIPPET.

Expectation templates were manually configured in Experiment 1, but that is infeasible for this larger rhythm space. Therefore we designed a simple procedure to configure a large set of expectation templates, approximating the enculturated expectations of a listener, through analysis of representative musical corpora (see Methods). Briefly, we derive the likelihood of three-interval rhythms from the relative frequency of event sequences within the metrical cycle of rhythms from a given music corpus. The likelihood associated with each three-interval rhythm is used to configure the prior probability of each expectation template, $p_0^m$. In this work we use notated corpora, i.e. quantized rhythms consisting solely of integer interval ratios, hence templates consist of three-interval rhythms with ratios spanning simple (e.g. 1:1:1) and complex (e.g. 7:2:3) patterns (see S4 Fig). This approach implements the hypothesis that musical features vary in frequency of occurrence across cultures [29, 30], resulting in different internalized rhythmic patterns and hence statistical affordances for meter [15].

We configured a single pPIPPET filter with expectations derived from a corpus of monophonic score-based German folksongs [46]. In order to illustrate the effect of prior template probability $p_0^m$ on pattern inference, we held the strengths and precisions of each template constant, and leave an investigation of the interaction between these terms in joint phase and pattern inference for future work. Fig 6A visualizes entropy of the pPIPPET filter's posterior distribution over expectation templates after being presented with each sampled three-interval rhythm (point in the rhythm space), using the `python-ternary` package [47]. For consistency with Experiment 3 we present each stimulus just once, but note that Desain & Honing [1] repeated each stimulus three times separated by one second gaps. Similarly to the entropy (consistency) of participant categorization in the study by Desain & Honing, areas of low entropy generally focus around rhythms related by the most simple integer ratios (1:1:1, and cyclic permutations of 1:1:2). Around these rhythms pPIPPET is able to infer an underlying rhythmic pattern (expectation template) with the lowest predictive uncertainty, suggesting these rhythms serve as strong perceptual attractors due to a high prior likelihood. In contrast, rhythms approximately related by mildly complex integer ratios (e.g. cyclic permutations of 2:2:3 and 2:3:3) have high entropy. However, even in these high entropy regions, Fig 6B reveals clear clusters where neighboring rhythms are categorized according to similar patterns related through simple integer ratios (integers less than 3). This suggests that the derived expectation templates are comprehensive in categorizing the rhythm space, albeit with varying levels of certainty.

Consistent with participant categorization in [1], the regions of low entropy form quasi-convex shapes that are delineated by regions of relatively high entropy, implying that some parts of the rhythm space are ambiguous. There are three possible explanations for these regions of high uncertainty in pPIPPET: (1) overlapping patterns of expectations, resulting in causal ambiguity; (2) lack of expectations for the pattern of observed events, i.e. observations are not explained by the generative model; and (3) weak prior expectations for the observed pattern, relative to other plausible patterns. Therefore, pPIPPET could be directly applied to resolve competing hypothetical explanations about the perception of rhythmic categories. One example includes the relationship between 2:1 and 4:3 prototypes for Malian participants, as simulated in Experiment 1. Polak et al. [5] proposed that these categories could be independent and (possibly) overlapping, or alternatively they might be (long-short) subcategories. In pPIPPET these hypotheses might be modeled using either separate or merged patterns of

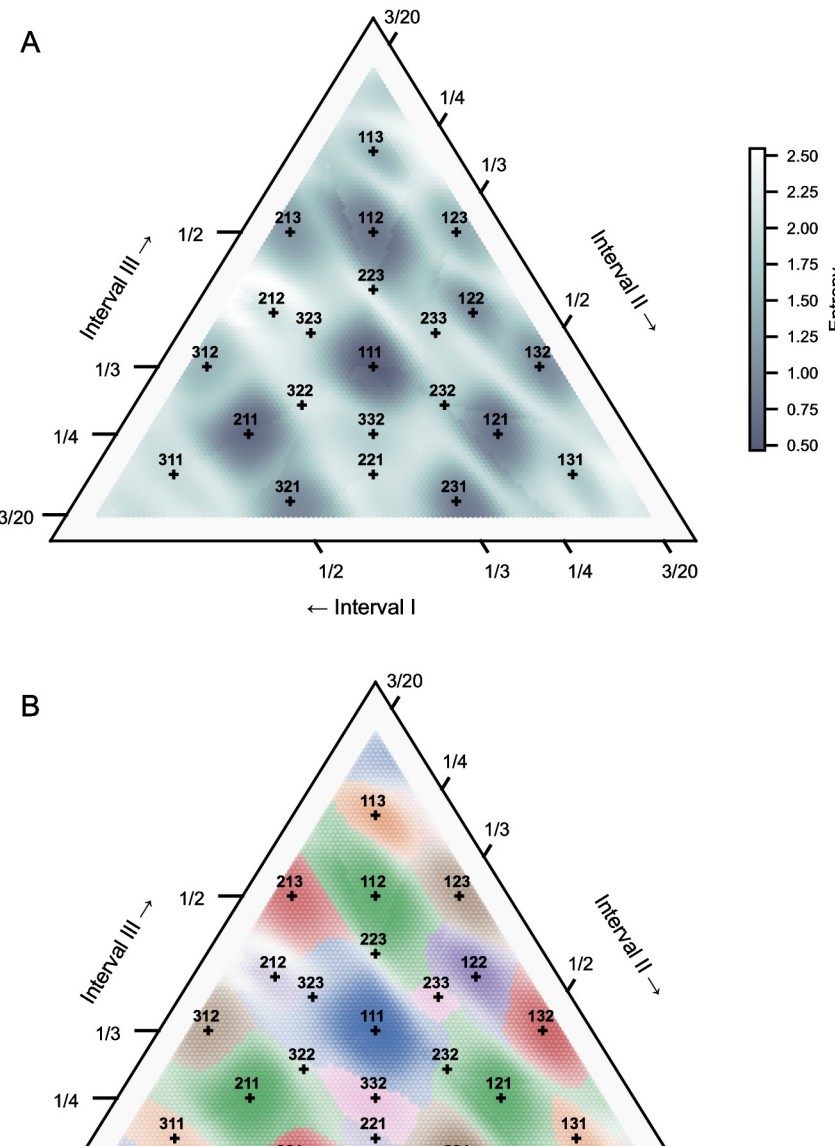

**Fig 6. Categorization maps for three-interval rhythms.** Categorization of three-interval rhythms (pattern duration of 2000ms and minimum interval duration of 300ms) after being presented once, using a pPIPPET filter configured with expectation templates derived from a corpus of German folksongs [46]. Rhythms related by small-integer-ratios (integers less than 3) are marked with crosses and labeled. A) Entropy of the posterior distribution over expectation templates. B) Colors correspond to cyclic permutations of the pattern which has been inferred (i.e. the expectation template which maximizes $p^m$), and transparency the entropy. For comparison, see [1, Fig 10b] and [1, Fig 13a]. This figure, and other ternary plots presented in this work, were made using the `python-ternary` package [47].

expectation, respectively, and application to intermediate stimuli (between 2:1 and 4:3) might prompt specific predictions for further experimental work. In Experiment 3, we examine categorization in further detail, focusing on its dependence on the enculturated expectations of a listener.

## Experiment 3: Iterated reproduction of non-uniform rhythms

The experimental paradigm used by [1] whose data was simulated in Experiment 2 relies on participants having sufficient musical training to be able to notate rhythmic stimuli. The iterated reproduction paradigm [24, 25] obviates this need, facilitating collection of empirical data from different musical cultures. Here, we simulate the iterated reproduction of non-uniform three-interval rhythms. In addition to implicit categorization of rhythms in a large continuous space, this complex task involves synchronization to highly unbalanced rhythms. As described in the Introduction, Jacoby et al. [24, 25] have used this paradigm to measure the perceptual priors (categories) on rhythm for listeners from different cultures. We simulate this data by comparing the implicit rhythm categorization of two pPIPPET filters, and relate the results to empirical observations. Alongside the filter created in Experiment 2 using monophonic German folksongs [46], we created a pPIPPET filter with expectations derived from monophonic Turkish makam music [48]. We refer to these filters as the German and Turkish models, but again stress that the music corpora used to train these models only approximate the listening experience of individuals from these musical cultures.

We directly simulate the method of Jacoby et al. [24, 25]. Trials involved five iterations of simulated tapping, each with 10 repetitions of the stimulus rhythm. For each pPIPPET filter we perform 2000 trials, using the same 2000 stimuli sampled uniformly from the rhythm space (Fig 3B). On the first repetition of each stimulus, we use the pPIPPET filter to determine the most plausible expectation template $m$, i.e. which maximizes $p^m$. Then, we use a PIPPET filter to track the remaining repetitions using the inferred template $m$. We separate these stages for computational efficiency (the tractability of pPIPPET is addressed in Methods), but note that participants in [25] were reported to spend roughly one repetition listening to the stimulus prior to synchronizing their tapping. After the first iteration, the stimulus rhythm is determined based on the phase tracking of the previous iteration. Specifically, events in the new stimulus are placed at the phase corresponding to the mean, across cycles in the previous stimulus, of the updated posterior distribution on phase at the time step of each event.

We hold the strengths and variances constant for the inference of a pattern, as in Experiment 2, but parameterize the variances and strengths when tracking the remaining repetitions. This introduces variability in the contribution of prior expectations to phase inference, depending on the rhythmic pattern (i.e. template $m$) inferred in a given trial. As per the scheme described in Methods , higher prior likelihoods of a template increase the strength $\lambda_i^m$ and precision $1/v_i^m$ of expectations. This assumes that exposure to specific rhythms in a musical culture leads to more precise expectations, affording accurate synchronization [49].

The distribution of rhythms in the final iteration are compared in Fig 7. We use kernel density estimates as in [24, 25] to compare the relative distribution of reproductions, which were biased away from the actual (uniformly sampled) stimulus rhythms. As in empirical data [25], the posterior on phase is systematically biased towards discrete modes at small-integer ratios (integers less than 3). This follows from Fig 6B, which revealed prominent rhythm categories at these ratios in this rhythm space, for the German model. The category weights for each model follow a similar distribution (Fig 7C), which is in-keeping with the similar distributions of simple onset patterns in the Turkish and German corpora used to configure the models (see Methods, and Fig 8). For example, the most prominent modes are at the simplest patterns for both models, isochrony (1:1:1) and 1:1:2 (and cyclic permutations), both of which appeared to be strong perceptual attractors in the entropy maps of the previous section (Fig 6).

However, there is notable variation between the models, with a significantly greater weight for the 2:2:3 category in the Turkish model ($p < 0.001$, Bonferroni-corrected). This is consistent with Jacoby et al.'s post-hoc analysis of empirical data, whereby the mode at the 2:2:3

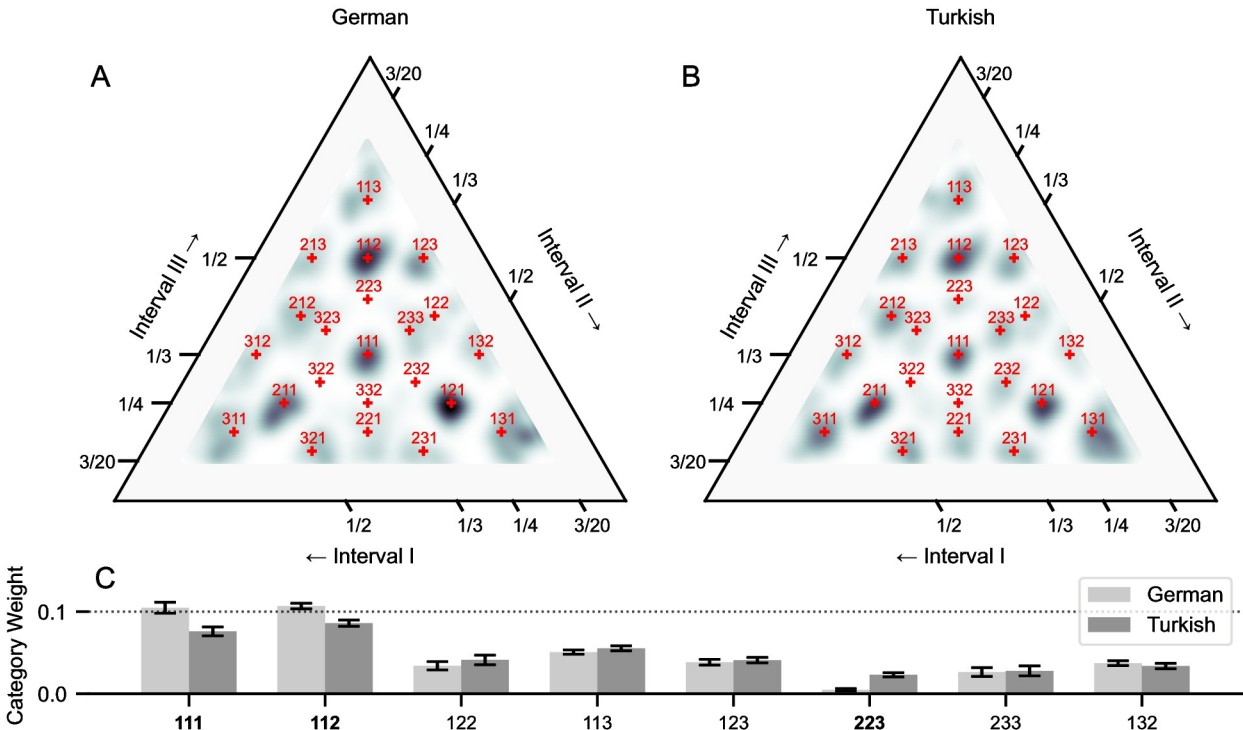

**Fig 7. Simulated iterated reproduction.** Results from the final iteration of all simulated trials, using pPIPPET filters configured with either German folk songs [46] or Turkish makam music [48]. A) Kernel density estimate (KDE) of the underlying data distribution for the German model, using the non-parametric method described in [24], normalized relative to a uniform distribution. B) KDE for the Turkish model. C) Category weights for the two models, obtained by fitting a constrained Gaussian mixture model (GMM), using the modified expectation-maximization procedure described in [25]. Weights for cyclic permutations of each ratio are grouped. Error bars reflect confidence intervals (SD of weights, derived from bootstrapping, N = 250). We draw specific attention to the differences in weights measured for cyclical permutations of the ratios labeled in bold (1:1:1, 1:1:2 and 2:2:3), which relate to differences in the KDE plots at the respective ratios.

rhythm was more pronounced for traditional musicians from regions where this rhythm is prominent in the local musical system, such as Turkey [25, p. 12]. Unlike the empirical data, here we can directly link the measured difference in 2:2:3 category weights to musical systems: metrical analysis of the Turkish corpus reveals a more uniformly distributed meter (i.e. flatter and less distinct metrical hierarchy) than the German corpus, consistent with the findings of a larger investigation of Turkish makam music [18]. This results in a greater prior probability and precision of expectation for this mildly complex interval pattern in the Turkish model. For the German model, rhythms approximately related by this ratio appear instead to be attributed to the neighboring 1:1:2 mode, which has a significantly greater weight for the German model ($p < 0.001$, Bonferroni-corrected).

There are several categories with relatively weak or strong modes for both models, in comparison to many groups in the empirical data: the 2:3:3 modes were weak, and the 1:1:1 mode was strong relative to the 1:1:2 modes. We address these differences in the Discussion, with respect to: the possible over (or under) representation of interval patterns in music corpora used to approximate encultured expectations; and differences between perceptual and motor entrainment.

## Discussion

We have presented pPIPPET, a model of entrainment to a time series of events, which draws upon prototypical patterns of temporal expectations in order to accurately track the event

stream. pPIPPET represents competing patterns of temporal expectations through point processes. The pPIPPET filter uses variational Bayes to continuously estimate the posterior probability that observed events are generated by each expectation template, whilst using these likelihoods to produce a marginalized estimate of phase and phase uncertainty. Put simply, pPIPPET is able to infer the pattern(s) best explaining an event stream, and uses these patterns to produce expectations for future events; whilst remaining open to the possibility of adjusting expectations when the stimulus rhythm changes.

pPIPPET provides a formal, quantitative characterization of how the enculturated expectations of listeners inform entrainment, reproducing cross-cultural variation in human entrainment when filters are configured with different discrete sets of expectations. We demonstrated the systematic bias incurred when tracking a patterned rhythm without a corresponding pattern of expectations [5]; and conversely the perceptual bias towards specific rhythmic patterns during iterated reproduction. In doing so, we related the implicit pattern inference in pPIPPET to empirical data on perceptual categorization of rhythms [1]. By configuring models' expectations according to the occurrences of rhythmic patterns in corpora of notated musical rhythms from different cultures, without constraints on the patterns that could be learned, models exhibited biases towards simple integer ratio rhythms alongside culture-specific deviations toward (mildly) complex integer ratios that distinguish musical cultures. These behaviors emerge naturally when entrainment is described as a process of Bayesian inference, PIPPET [9], where precise phase-based temporal expectations are optimized using prior expectations for different rhythmic patterns.

## Simulating enculturated expectations

When perceptual or motor entrainment is cast as an inference problem, pPIPPET exposes several interpretable parameters that can be used to simulate the enculturated bias observed in rhythm perception and production experiments. Here, we varied: (1) the discrete set of rhythmic patterns represented by expectation templates; (2) the prior probability $p_0^m$ that events were generated by a given template $\lambda^m$; and (3) the expectation strength $\lambda_i^m$ and (inverse) expectation precision $v_i^m$ associated with events in each template. Both (1) and (2) influence the ability to correctly infer a pattern underlying an event stream, where tracking a rhythm using an inaccurate (non-matching) pattern of expectations causes systematically biased phase estimates. Then (3) influences the strength of error correction upon event observations, proportionally to the posterior probability $p^m$ of the respective template. We leave a thorough exploration of the interaction between expectation precision and template probability to future work; noting that the results presented do not essentially depend on the configuration choices made.

For a hypothetical listener from a given musical culture, we configured pPIPPET parameters by analyzing metrical patterns in symbolic musical rhythms which characterize their musical experience [15], assuming the relative frequency of patterns determines the precision of and prior probability associated with corresponding internal models (expectation templates). This approach is consistent with statistical learning descriptions of stylistic enculturation, whereby precise expectations are acquired for commonly observed rhythmic patterns in a musical culture [14, 16], enabling precise synchronization for these culturally-familiar rhythms [49]. We have not explicitly distinguished between effects of implicit statistical learning and explicit musical training, though the enculturated biases qualitatively reproduced in Experiment 1 and Experiment 3 were reported empirically for trained musicians. Carefully designed experiments in future work will help to tease apart exactly how expectation strength $\lambda_i$ and

precision $\nu_i$ in pPIPPET respectively relate to prior expectations in entrainment, alongside effects relating to explicit musical training [7].

When two pPIPPET filters are configured with expectations derived from different musical cultures, the difference in entrainment quality when applied to each musical style can be related to the *cultural distance hypothesis*—the degree to which music from two cultures differ in statistical patterns of rhythm will predict how well someone synchronizes to music from the other [50]. Cultural distance has been modeled as differences in information-content between two models of auditory expectation [51], each using statistical learning to learn a probabilistic grammar characterizing a musical style (both in terms of pitch and rhythm) [16, 21]. These models have been shown to predict Western listeners' expectations in several experimental paradigms (see [16, p. 384] for a review), but have not been applied to empirical research demonstrating cultural variation in expectation and uncertainty. Given that pPIPPET explicitly describes entrainment, it can readily be applied to empirical research involving rhythm production, a task well-suited to cross-cultural research due to the lack of language and musical notation [52].

In this work we derived precise phase-based expectations from discretized (symbolic) interval patterns, as notated in sheet music, allowing us to derive highly stereotyped representations of rhythmic patterns. This approach is limited in that relatively few non-Western music corpora are available in a computational notated form [22, 23], and typically a Western-trained musical expert is required to notate and/or validate such corpora, which might introduce biases. However, there is no reason why pPIPPET templates need to be configured with expected events related by perfectly discretized intervals. A non-trivial extension of this work would involve configuring pPIPPET according to analysis of performed musical rhythms, i.e. non-quantized rhythms featuring non-integer interval ratios. This might be tackled by identifying concentrations in the probability density of specific interval patterns as in [53], or inferring latent hierarchical structure from event timing (e.g. [54]). Given a sufficiently large training corpus of performed music, it would be interesting to evaluate whether a model configured this way would reproduce similar integer ratio biases as in the present work.

Extracting templates from performed music would also provide a natural opportunity to consider the influence of tempo (rate of phase advance). This would be a non-trivial extension of this work, insofar as tempo appears to influence the categories of rhythm observed in musical rhythms [53, 55, 56], possibly reflecting an interaction with meter perception [20]. Polak et al. [5] emphasize that the reliable reproduction of a 4:3 rhythm by Malian drummers occurred at a fast tempo, an important contextual factor for their performance practice, but this was not included in our description of the behavior in Experiment 1. This might be explored using PATIPPET [9], a variant of PIPPET which infers tempo as a (dynamic) hidden variable governing the rate of phase advancement.

Future work should scrutinize which interval patterns are overly (or under) represented in music corpora used to approximate enculturated expectations. For example, long-short interval patterns occur more frequently in Western musical scores than short-long permutations, yet there is limited evidence that variability in synchronized tapping is lower for long-short patterns [55, 56]. Further, Jacoby et al. [25] reported a slight tendency for reproduction of a rhythm's cylic permutation where the final interval is the longest. One approach to address this in pPIPPET would be re-weighting priors derived from music corpora to reflect the asymmetries demonstrated empirically (see 'optimal priors' in [2]). It is also possible that some cultural differences in the perceptual grouping of rhythms (e.g. into either long-short or short-long patterns) might not reflect musical patterns at all, but features of a listener's native language, such as prosodic stress patterns [57].

Finally, given that PIPPET is formulated as a perceptual process, we cannot ignore the possibility that differences between empirical iterated reproduction results and those of

Experiment 3 (using pPIPPET) stem from a distinction between perceptual and motor entrainment. However, we expect that the same phase inference processes underpin both perceptual and motor entrainment. Experimental work such as [58] has shown that perceptual expectations for rhythmic patterns directly influence entrained movement. Additionally, in [59], two-interval rhythms (1:3 to 1:7) were biased towards more even ratios in both perception and production tasks, instead of performed rhythms compensating for a perceptual bias. Physical entrainment might therefore be approached as a series of constraints (e.g. for specific motor effectors) on top of perceptual entrainment. An additional consideration is the contribution of sensory (e.g. proprioceptive, tactile) feedback from movement on phase estimates (e.g. [60, 61]). This might be modeled by extending pPIPPET's generative model to infer a single underlying phase from multiple types of events (as in mPIPPET [9, p. 8]), using a template to represent the alignment between expected sensory feedback and phase.

## Intrinsic rhythmic preferences

The phase estimation mechanism in PIPPET, and by extension pPIPPET, has no intrinsic bias for the inference of certain rhythmic patterns—notably, those related by small integer ratios. By contrast, the universal tendency to perceive and (re)produce rhythms related by small integer ratios [31] raises the possibility of physiological and/or cognitive constraints on rhythmic priors (as considered in [24, 25]). We have shown that present empirical results can be accounted for as a process of probabilistic learning from real-world notated musical rhythms (featuring both simple and complex integer interval ratios). Presumably, musical systems serve as a proxy for priors if they reflect perceptual biases and production constraints [62]. However, we don't discount the possibility of intrinsic (innate) priors which could be incorporated into the model.

The phase inference framework describes computational principles underlying expectancy-based entrainment, and does not commit to a precise mechanistic interpretation of entrainment [8]. If constraints in a specific mechanistic model limit the learning of rhythms we experience, e.g. to small integer ratio rhythms, then pPIPPET might be refined by starting with certain priors as a substitute for these constraints; or, some appropriate basis functions might be defined for the generative model. This would *not* fundamentally change pPIPPET, but constrain the rhythms that could be represented in expectation templates. But it is clear that with appropriate music-cultural experience, people can form expectations flexibly for specific and complex rhythms, including rhythms that are not easily quantizable into integer-ratio intervals (e.g. [45, 63]); and this extrinsic aspect of learning might still be amenable to a probabilistic description. The challenge for future experimental work will be to empirically distinguish between model configurations with extrinsic or (partially) intrinsic priors.

## Relationship to other models

Further to existing models in the phase inference framework [9], pPIPPET is able to infer which of multiple expectation templates optimize ongoing phase inference, such that phase corrections depend on the specific (enculturated) set of expectations configured. In contrast, existing probabilistic models of enculturated expectations are limited to discretized (symbolic) representations of intervals [15]—as is common in probabilistic models (e.g. [13, 17, 18])—so are not structured to infer phase in continuous time. Instead, these models address how temporal expectations are learned: estimating parameters of a generative sequential model through statistical learning [16], given an empirical sample of symbolic rhythms. Therefore they might form the basis of a more sophisticated approach to derive expectation templates in pPIPPET,

given that they benefit from context-dependent predictions, and can incorporate explicit hypotheses for meter [14] (metrical inference with pPIPPET is considered in the next section).

Bayesian inference has already been used to predict categorization of non-uniform rhythms [2] for the empirical data explored in Experiment 2, though not as a model of real-time behavior. This model also used priors derived from music corpora, but interestingly combined these with continuous production data—an approach that might be considered in deriving expectation templates. Recurrent attractor networks for rhythm quantization have also been applied to rhythm categorization [64, 65], configured with polynomial attractor functions resembling the repertoire of expectation templates in pPIPPET, but it is unclear how these models might be applied to real-time behavior.

Neural oscillator models which use Hebbian plasticity [37] have simulated effects of musical experience on entrainment dynamics [38, 39], and might be capable of simulating the empirical results addressed here with pPIPPET. The computational description of enculturated biases in pPIPPET–which depends on inferring the state of an underlying cyclical process–might be compatible with a range of mechanistic implementations, including oscillatory processes. Hence, we do not believe these approaches are mutually exclusive. Instead, they might be considered to approach the problem from different directions: bottom-up, respecting the dynamics of cortical oscillation at the physiological level; or top-down, based on (possibly Bayesian) computations explaining the breadth of empirically observed entrainment behaviors. In other words, as recently proposed in [66], oscillator-based models could be viewed as physiological mechanisms underlying rhythmic inferences that can be described computationally using Bayesian principles. Another interesting point of convergence is in the experience-dependent learning mechanisms proposed—implicit statistical learning might follow the very associative principles of Hebbian learning [67], and Hebbian plasticity has been directly related to parameter optimization in hierarchical generative models used to describe cortical organization (e.g. [40, p. 824]).

## Relating pattern inference to metrical inference

Behavioral studies examining phenomena such as categorical rhythm perception through entrainment are limited by practical constraints to studying two- or three-interval rhythmic patterns. This is a microcosm of rhythm perception, making it unclear how the perceptual biases observed in these tasks relate to features of naturalistic musical rhythms, in particular *metrical structure* (a hierarchically embedded set of time periods inferred from and aligned to a rhythm). In the present work, we derived expectation templates from empirical samples of musical rhythms, according to the relative likelihood of different onset patterns within a metrical cycle (i.e. metrical phase). Therefore the "patterns" represented in expectation templates correspond to prominent metrical patterns in the respective music corpus. The perceptual biases of simulated iterated reproduction in Experiment 3 therefore stem from metrical priors, in keeping with Jacoby et al.'s speculation that measured priors in empirical iterated reproduction reflect expected (metrical) groupings of rhythmic events [25, p. 17].

Considering naturalistic rhythms, pPIPPET could be applied—in exactly the same way as in the present work—to infer the pattern of metrical beats induced by a stimulus given some prior expectations, hence providing a real-time simulation of metrical inference. This is possible because there is no requirement for a one-to-one relationship between rhythmic events and expectations in PIPPET, such that it can describe how metrical beats (which might span more than one rhythmic event) are tracked within an event stream [9]. Further, pPIPPET could be used to model enculturation effects on inferred metrical structure. Lenc et al. [68] recently proposed this as the most sophisticated form of metrical perception, stemming from

## Schematic and continuous temporal expectations

pPIPPET's discrete set of expectation templates are represented as continuous functions of phase (Eq 4), which might even overlap if phase-based onset times $\phi_i^m$ are sufficiently close and $\nu_i^m$ sufficiently large. Many models of rhythmic grammar are limited to symbolic representations [14, 17, 70, 71] which can be related to schematic expectations [72], abstract temporal representations learned through extensive exposure to music and activated by appropriate musical contexts. Given that symbolic rhythms can be used to construct expectation templates, pPIPPET can be used to investigate how events with continuous timing variation are resolved against schematic expectations. This behavior is similar to models of categorical rhythm perception that perform quantization, analyzing performed rhythms with continuous timing variations, and identifying discretized representations of intervals (related by integers) suitable for music transcription [64, 65, 73, 74].

Classic quantization models have been criticized for separating and disregarding the 'non-categorical' component of an identified pattern or meter [75] (also see [76]), which some argue corresponds to the microtiming deviations which characterize expression in a musical performance [77]. In the context of PIPPET, timing deviations between observed events and expectations are not just discarded as noise, but are integrated into the posterior estimate on phase, contributing toward optimal tracking of a rhythm (similarly to event-based error correction models in sensorimotor synchronization [78, p. 407]). Specific patterns of prediction error or phase uncertainty in pPIPPET might therefore be related to qualitative timing "feels" (e.g. "pushed" or "laid-back" timing), stemming from unique probability density functions of timing in different performances of a rhythm [79], with respect to metronomic expectations. However, it is also possible that schematic expectations are shaped by stable timing patterns, hence forming qualitatively distinct meters; as London notes, "characteristic timings will become part of our habitual entrainments to them" [20, p. 154]. Consistently, given that PIPPET is not tied to quantized representations of rhythms as found in Western scores, this allows us to posit that any rhythm which can be highly practiced through dance or music performance (or even listening) can become a unique template for event expectation, even if it is not easily quantizable into discrete integer-ratio intervals. This is particularly relevant to musical styles where there isn't a clear divide between rhythmic structure and expressive timing patterns [44, 63, 80]. One popular example of this is the 'swing' feel in jazz, where the central rhythmic quality stems from an uneven performance of rhythmic events notated with even duration (eighth notes); such as the distinctive 'ding-ding-a-ding' rhythm often performed on the ride cymbal [81]. Therefore, if different timing patterns (e.g. senses of swing in different jazz styles) were represented as distinct patterns of expectation in pPIPPET, then pattern inference might be related to the recognition of qualitatively distinct metrical timing patterns, as described in London's "many meters hypothesis" [20, p. 154].

## pPIPPET in the brain

PIPPET, which serves as a basis for pPIPPET, is an abstract cognitive model that does not commit to a specific brain-based implementation. However, Cannon proposed an approximation of PIPPET in the brain [9], and we can develop this description by considering where the pattern inference mechanism of pPIPPET would extend a neural implementation of PIPPET's phase estimation.

The proposed implementation of PIPPET draws upon the hypothesis that simulated actions in motor planning regions provide temporal predictions about auditory events. There is a vast literature highlighting a central role of the motor system in rhythm perception [82–84], but this most directly draws upon Patel & Iversen's "Action Simulation for Auditory Prediction" (ASAP) hypothesis [85]. A recently updated account of ASAP [86] proposed that the dorsal striatum orchestrates supplementary motor area (SMA) dynamics, allowing precisely patterned time-keeping for the anticipation of repeating rhythmic patterns. In the context of PIPPET, estimated phase and phase uncertainty are hypothesized to be represented in medial premotor cortex (MPC), while basal ganglia (BG) selects an expectation template based on the recent rhythmic context. This template is combined with the posterior estimate on phase in MPC to calculate a subjective hazard rate (level of anticipation), which is projected to the parietal cortex as a descending prediction for auditory events, as in hierarchical predictive processing [87].

We hypothesize that the pattern inference mechanism introduced in pPIPPET can be related to the recruitment of higher-order cognitive mechanisms in prefrontal cortex (PFC). This would provide persistence of state and integration of information over longer time scales, enabling the inference of expectation template plausibility across successive auditory events, and informing selection of expectation templates in BG. It has been proposed that prefrontal regions support selection of rhythm representations [88], and indeed PFC activation appears to increase when listening to metrical and non-metrical rhythms (i.e. musical beat selection) in contrast to simple isochrony [89]. Similarly, data has shown that PFC is engaged in musicians during rhythm encoding and retrieval, while less so during rhythm maintenance [90]. PFC activity is also greater in musicians than non-musicians during a tapping task [91], the authors arguing that this relates to superior ability to extract temporal features relating to rhythmic structure, with the PFC mediating working memory. Within the working memory system, one region of interest might be ventrolateral PFC (VLPFC). VLPFC appears to be engaged for active memory retrieval tasks requiring top-down control and selection between options [92–94], and activity has been shown to correlate with BG activity during synchronized tapping to auditory rhythms [95], increasing for rhythms with greater metrical complexity. More generally, many studies have implicated the inferior frontal gyrus (IFG) in temporal hierarchy processing, as highlighted by a recent meta-analysis of neuroimaging studies investigating musical rhythm and linguistic syntax [96] (also see [97]).

One main difference between our formulation of pPIPPET and its neural implementation might be an unrealistic degree of parallelism, given that likelihood estimates for all expectation templates are concurrently updated on every time step (tractability is discussed further in Methods). pPIPPET's pattern inference might therefore reflect a specific mode of rhythm perception that is only activated to identify a pattern of temporal invariances (as in Experiment 3), or once a previously-identified pattern incurs significant errors in phase estimation. In other words, it might reflect rhythm recognition as opposed to rhythm continuation [20, p. 50]. This aspect of pPIPPET could be tested using an experiment requiring participants to "re-hear" a rhythm due to structural transitions, such as increased syncopation [98, 99] or metrical modulations [3]. In these cases, we hypothesize that pPIPPET would offer a distinctive explanation for bistable rhythm percepts on transitions: simultaneous phase tracking and pattern inference would (at least initially) adjust predicted phase in the favor of a template with high likelihood ($p^m$), while evidence for a new and better-matching template accrues. Additionally, in the context of reproduction tasks (such as in Experiment 3), we hypothesize that longer periods of pattern inference would allow accurate expectations with low prior likelihoods to converge on higher posterior likelihoods (i.e. delayed recognition of the stimulus pattern), which would be reflected in changing patterns of tapping asynchrony within a trial.

### Conclusions and future considerations

In this paper we presented and evaluated pPIPPET, which provides a plausible account of the cognitive mechanisms underlying effects of enculturation on rhythm perception and production demonstrated here through simulations of existing empirical data covering listeners from several musical cultures.

Our results support a plausible mechanism for the process by which enculturated enculturated expectations bias entrainment, involving an approximation to optimal (Bayesian) inference about the rhythmic stimulus using prior expectations for rhythmic patterns learned within a given musical culture. The present simulations were limited to configuring pPIPPET using (real-world) notated musical rhythms containing both simple and complex integer-ratio rhythms. A challenge in future work will be to replicate our simulations with pPIPPET learning from performed rhythms, requiring careful consideration of how enculturated temporal expectations are extracted from rhythms with non-integer ratio rhythms and variable tempo. More detailed simulations of specific empirical studies will also enable exhaustive analysis of the interaction between expectation precision and template probability in pPIPPET.

The present work is most obviously relevant to understanding music perception. However, the model can be applied in several domains that involve tracking rhythm. As pPIPPET is able to account for non-isochronous entrainment, it might be particularly well-suited to tracking aperiodic patterns in speech, which some argue poses challenges for oscillatory models [100] in the absence of top-down content-based predictions [101]. Future work might also explore "veridical" expectations [72], activated by memory traces acquired through short-term learning—neural and behavioral evidence suggests that "memory-based" expectations rely on partly different underlying mechanisms to traditional "beat-based" entrainment [102]. Similarly to "multiple-viewpoint" systems that maintain both short- and long-term probabilistic models of rhythmic structure and combine their respective predictions [16, 103], pPIPPET could be extended to incorporate expectations relating to learning over different timescales.

## Methods

In this work we are concerned with how temporal expectations, as shaped by long-term music-cultural exposure, dynamically inform entrainment—*expectancy-based entrainment*. The role of temporal expectations in rhythm perception and entrainment can be understood through the predictive processing framework [84, 104–106], whereby the brain continuously infers the hidden causes of sensory events by using learned anticipatory models of temporal regularities.

Several models of expectancy-based entrainment have been described as formal inference problems, with precise solutions, within Cannon's "phase inference framework" [9]. Centrally, "Phase Inference from Point Process Event Timing" (PIPPET) describes how an observer continuously infers phase—a hidden variable that advances over time with some amount of random noise—from a sequence of events, such as the phase of a beat cycle in a musical rhythm. Event generation is modeled as a point process, and described through a generative model that is variationally inverted for use in a continuous inference process. The filtering equations that approximate an optimal Bayesian solution to this problem are presented in the next section, and the derivation is presented in S2 Text.

Following the same general approach, we formulate an additional version of this problem, pPIPPET, which generalizes PIPPET to incorporate competing event-based expectation patterns, each with event expectations at a distinct set of phases with distinct degrees of expectation precision—in other words, unique expectations for different plausible rhythmic patterns, each of which is assigned a likelihood that updates continuously alongside phase inference.

Again, the solution is presented, but please refer to S2 Text for the derivation. Finally, we describe how these competing expectations can be derived from music corpora, as in this work.

## PIPPET: Phase inference from point process event timing

**Problem.** Given the timing of an event sequence, generated as a point process whose rate is characterized by a function of phase, PIPPET reflects the problem of continuously estimating an underlying (hidden) noisy phase variable, $\phi \in \mathbb{R}$.

A generative model of rhythmic events can be described using a drift-diffusion process $\phi$ to represent rhythm phase, and an inhomogenous point process $\lambda(\phi)$ for event times, that is modulated by phase. The inhomogeneous point process describes event generation through a probability function $\lambda(\phi)$. Referred to as an "expectation template", $\lambda(\phi)$ reflects the observer's expectations for events at specific phase values. It is composed of a summation of Gaussian distributions ($\varphi$), indexed by $i = 1, 2, \cdots N$, each centered at a mean phase $\phi_i$, with variance $v_i$ and scale $\lambda_i$:

$$\lambda(\phi) = \lambda_0 + \sum_{i}^{N} \lambda_i \, \varphi(\phi|\phi_i, v_i) \tag{1}$$

where:

- $\phi_i$ is the mean phase of an expected event.

- $v_i^{-1}$ is the temporal precision of an expected event.

- $\lambda_i$ is the strength of expectation for an event.

- $\lambda_0$ is the rate of events being generated by uniform background noise.

The expectation template $\lambda$ flexibly describes a stochastic sequence of events, which need not be periodic in nature, such that $\phi$ should be thought to advance along the real number line as opposed to a circle (see Fig 2).

The goal of the observer is to infer a posterior distribution describing the estimate of phase $\phi$ at time $t$, $p_t(\phi) = p(\phi|N_{\tau<t})$, for a sequence of event times $\{t_n\}$. At time $t = 0$ this is initialized with a prior distribution $p_0(\phi)$, and is updated using the expectation template $\lambda(\phi)$ and an event-counting function $N_t : \mathbb{R} \mapsto \mathbb{Z}^{0+}$.

Through variational inference, the true but intractable posterior distribution can be approximated in Gaussian form (the Laplace approximation), by selecting the mean and variance which minimize the Kullback–Leibler divergence from the true (optimal) posterior at each $dt$ time step. Cannon's solution is a generalized Kalman-Bucy filter with Poisson observation noise [9].

**Solution.** In the solution proposed by Cannon [9], the Gaussian posterior updates continuously between events at every time step, and additionally updates after event observations. At any time $t$, the sufficient statistics are denoted by the mean $\mu_t$ and variance $V_t$. If an event occurs at time $t$, these values (left-hand limits) are revised to $\mu_{t+}$ and $V_{t+}$ (right-hand limits).

On each time step, $\mu_t$ and $V_t$ evolve according to the stochastic differential equation below:

$$\begin{cases} d\mu = & dt + (\hat{\mu} - \mu_t)(dN_t - \Lambda dt) \\ dV = & \sigma^2 dt + (\hat{V} - V_t)(dN_t - \Lambda dt) \end{cases} \tag{2}$$

Then, if an event occurs at time $t$, $\mu_t$ is updated to $\mu_{t+}$, and this is used to update $V_t$ to $V_{t+}$ as specified below:

$$\mu_{t+} = \hat{\mu} := \frac{\lambda_0}{\Lambda}\mu_t + \sum_{i=1,\dots}\frac{\Lambda_i}{\Lambda}\hat{\mu}_i$$

$$V_{t+} = \hat{V} := \frac{\lambda_0}{\Lambda}\left(V_t + (\mu_t - \mu_{t+})^2\right) + \sum_{i=1,\dots}\frac{\Lambda_i}{\Lambda}\left(\hat{V}_i + (\hat{\mu}_i - \mu_{t+})^2\right)$$

where

$$\hat{\mu}_i := \frac{V_t^{-1}\mu_t + v_i^{-1}\phi_i}{V_t^{-1} + v_t^{-1}} \quad \text{and} \quad \hat{V}_i := \frac{1}{V_t^{-1} + v_i^{-1}}$$

$$\Lambda_i := \lambda_i\,\varphi(\mu_t|\phi_i, v_i + V_t) \quad \text{and} \quad \Lambda := \sum_i \Lambda_i$$

(3)

In these expressions,

- $\Lambda$ is the extent to which an event is anticipated at time $t$ given the level of phase uncertainty, referred to as the "subjective hazard rate". $\Lambda_i$ is the anticipation related to the event expectation with index $i$, the "conditional subjective hazard rate".

- $\hat{\mu}_i$ is a precision-weighted sum of the mean estimated phase $\mu_t$ and mean expected phase $\phi_i$ for the expectation $i$.

At events, Eq (3), the posterior on phase is reset to the Gaussian with mean $\hat{\mu}$ and variance $\hat{V}$. Notably, the variance $\hat{V}$ will increase if there are several candidate expectations that the observed event might correspond to. The variance will similarly increase if the background rate $\lambda_0$ and expectations are comparably likely. Between events, Eq (2), $\mu_t$ increases at a fixed rate on each time step $dt$, alongside accumulation of phase uncertainty as $V_t$ grows with rate $\sigma^2$.

In addition to the original formulation of PIPPET, two types of noise are introduced: (1) the timing of stimulus events are perturbed by event noise, $\eta_e$, which represent noisy delays in auditory processing; and (2) the estimated stimulus mean is perturbed by phase tracking noise, $\eta_\mu$, reflecting the noisiness of timekeeping processes in the brain. Crucially, these (constant) noise terms are *not* parameters of the observer's generative model, but instead properties of the physiological implementation of PIPPET. Event times $\{t_n\}$ are perturbed by noise drawn a standard Normal distribution and scaled by a constant $\eta_e\,\mathcal{N}(0,1)$, and the $d\mu$ term is perturbed by noise drawn a standard Normal distribution and scaled by a separate constant $\eta_\mu\,\sqrt{dt}\,\mathcal{N}(0,1)$.

## pPIPPET: PIPPET with pattern inference

**Problem.** pPIPPET generalizes PIPPET to incorporate multiple expectation templates, indexed by $m$, each of which corresponds to expectations for a specific rhythmic pattern. The observer is again tasked with estimating a (single) underlying phase $\phi$. In this case, they assume that the stimuli are generated by one of multiple expectation templates—template $m$ is chosen with probability $p^m$, and a stimulus generated by template $m$ is generated as a point processes with rate $\lambda^m(\phi)$:

$$\lambda^m(\phi) = \lambda_0^m + \sum_i \lambda_i^m\,\varphi(\phi|\phi_i^m, v_i^m)$$

(4)

At time $t$, the observer infers a distribution $p_t^m$ over possible generating templates $m$. This distribution starts with the prior distribution $p_0^m$ and is updated based on event observations at each $dt$ time step. This can be considered an implicit inference of the rhythmic pattern(s) underlying the observed events, and allows the observer to optimize phase inference given the relevance of each template $m$.

**Solution.** At any time $t$, the probability associated with each template $m$ is updated according to the ratio between its respective subjective hazard rate $\Lambda^m$ and the total subjective hazard rate $\Lambda$:

$$dp^m = p^m \left( \frac{\Lambda^m}{\Lambda} - 1 \right) (dN_t - \Lambda dt) \tag{5}$$

The posterior on phase, with mean $\hat{\mu}$ and variance $\hat{V}$, is a Gaussian approximation of the sum of (Gaussian) estimates made by each template $m$, weighted by their probabilities $p^m$:

$$\Lambda = \sum_m p^m \, \Lambda^m$$

$$\hat{\mu} = \sum_m p^m \, \hat{\mu}^m \tag{6}$$

$$\hat{V} = \sum_m p^m \hat{V}^m + \sum_m p^m (1 - p^m)(\hat{\mu}^m)^2 - \sum_{m \neq n} 2 p^m p^n \hat{\mu}^m \hat{\mu}^n$$

$\Lambda^m$, $\hat{\mu}^m$ and $\hat{V}^m$ are calculated as per Eqs (2) and (3), using the single (weighted) posterior from the previous time step as a prior, in the context of the respective likelihood function $\lambda^m(\phi)$. In other words, these are candidate posteriors, in the case that the events are being generated by template $m$.

The extent to which the likelihood of a template $p^m$ evolves at any time $t$ depends on the template's subjective hazard rate $\Lambda^m$ compared to other templates. If an event is observed at an unexpected time according to template $m$, i.e. $\Lambda^m$ is low, then the likelihood $p_t^m$ of template $m$ will decrease. This decrease will be larger if other templates strongly expect an event, as the $(\Lambda^m/\Lambda - 1)$ term will decrease. In the case that an event is equally predicted by all templates, the likelihood of each template will not change.

## Deriving expectation templates from music corpora

In order to simulate rhythmic entrainment—as formulated through pPIPPET—for an enculturated listener, a discrete set of expectation templates $\{\lambda^m(\phi)\}$ must be derived which reflect the expectations that emerge in listeners through (presumably) life-long learning. Assuming that musical systems incorporate production constraints [62] and innate perceptual biases, music corpora should offer reasonable approximations of a listener's discrete rhythmic priors (set of expected rhythmic patterns).

Here we derive templates from metrical analysis of music corpora: the probabilities by which event onsets fall on specific metrical positions, for each metrical category (time signature) within the corpus. These structures have been likened to schematic patterns of metrical salience which are acquired over long-term exposure to musical rhythms [17, 107], i.e. the hierarchical groupings of intervals characterising specific musical styles. Strong priors for rhythmic patterns associated with specific musical features have been observed in iterated reproduction tasks for participants with relevant musical experience [24, 25], suggesting exposure to specific rhythms—as represented in a given music corpus—facilitates accurate reproduction [49].

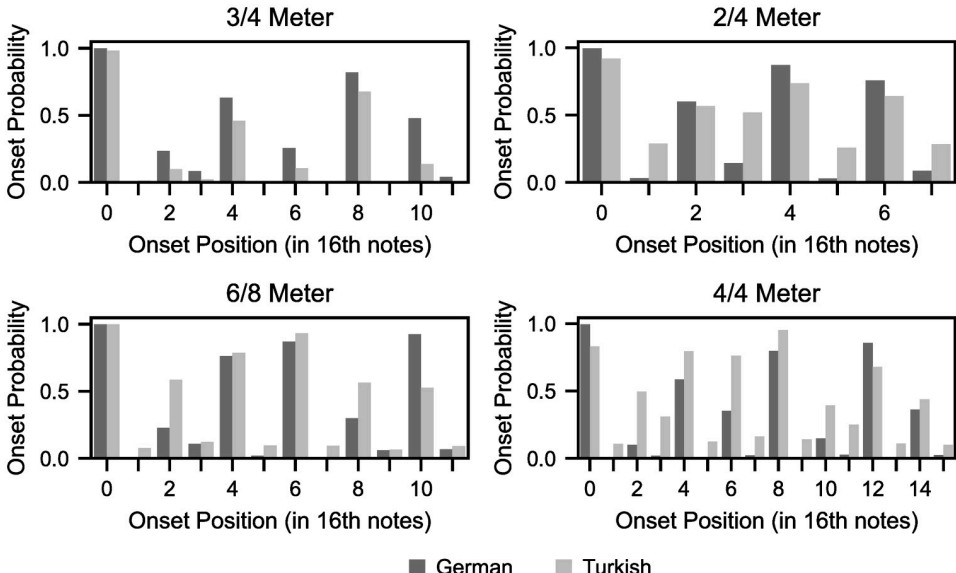

**Fig 8. Metrical analysis for German and Turkish corpora.** Relative frequency of onset positions (phase, at the resolution of 16th notes) within a metrical cycle of rhythms belonging to a specific meter (time signature). This analysis from [15, p. 185] is used with permission.

**Metrical models.**   We leverage the metrical analysis by Van der Weij [15], which compares the statistical properties of monophonic score-based rhythms in German folk melodies [46] and Turkish makam music [48]. The rhythm samples analyzed were carefully curated to ensure an equal number of total rhythms, which had been truncated to segments of uniform length, and filtered to only include rhythms defined at a sixteenth-note resolution. Each sample consisted of 79 rhythms in the following meters (time signatures): 2/4 and 4/4, binary simple meters; 6/8, binary compound meter; and 3/4, ternary simple meter. The metrical contexts analysed included *metrical phase*, the positions of onsets within the metrical cycle. Fig 8 shows the relative likelihood that onsets occur at different metrical phases, for each rhythm sample. While the pattern of onset probabilities is similar across the German and Turkish rhythms, there is greater uncertainty for the Turkish rhythms. This is consistent with a detailed analysis of Turkish makam rhythms that revealed a less stratified meter than in Eurogenetic rhythms [18].

**Expected patterns.**   For both the German and Turkish rhythm samples described above, the relative frequencies by which events occur at specific metrical phases are used to construct two sets of expectation templates. As the rhythm spaces explored in simulation consist of three-interval patterns, each template also reflects an expected three-interval pattern, but is derived from interval patterns within a metrical cycle.

Let $\mathcal{M}$ denote the set of meters, where each meter $M \in \mathcal{M}$ has a period $N_M$. First, we identify all possible three-interval patterns in each meter $M$ by the metrical positions $\langle n_i, n_j, n_k \rangle$ whereby events can occur. These patterns are not constrained in duration to a metrical period (an entire metrical cycle), and therefore can reflect beat groupings within the meter at different levels of periodicity. Next, we convert each of these patterns into a pair of ratios relating the consecutive intervals: $\langle r_1, r_2 \rangle$. Per rhythmic sample, the likelihood of each pattern is defined by the product of independent probabilities for each respective onset in a given meter $M$, summed

across $\mathcal{M}$ and normalized:

$$
\begin{aligned}
P(\langle r_1, r_2 \rangle) &= \frac{1}{|\mathcal{M}|} \sum_{M \in \mathcal{M}} P(\langle r_1, r_2 \rangle \mid M) \\
&= \frac{1}{|\mathcal{M}|} \sum_{M \in \mathcal{M}} P(n_i \mid M) \, P(n_j \mid M) \, P(n_k \mid M)
\end{aligned}
\tag{7}
$$

Finally, given a total absolute duration $t_d$ for three-interval stimulus rhythms, each expected ratio pattern $\langle r_1, r_2 \rangle$ can be converted into expectations for three intervals in time $\langle d_1, d_2, d_3 \rangle$, as per below. Each of these interval patterns characterize the expected phase of events within a template $m$, $\phi_i^m$. The prior likelihood of events being generated by this template $p^m$ is set to the probability of the respective ratio $P(\langle r_1, r_2 \rangle)$. Note that these templates are filtered to ensure the minimum and maximum expected durations conform to the rhythm spaces defined in the methods being simulated.

$$
d_3 = \frac{t_d}{\left( \frac{1}{r_1 r_2} + \frac{1}{r_2} + 1 \right)} \;, \quad d_2 = \frac{d_3}{r_2} \;, \quad d_1 = \frac{d_2}{r_1}
\tag{8}
$$

**Template parameters.** The expectation strength $\lambda_i^m$ and (inverse) expectation precision $v_i^m$ for the event expectations on each template $m$ are derived from the prior likelihood of events being generated by this template $p_0^m$. For now, we assume these parameters are held constant for all events within a template $m$. Given a maximum expectation strength $\lambda^{max}$ and maximum expectation variance $v^{max}$, we assign the template parameters by $\lambda^m = p_0^m \lambda^{max}$ and $V^m = (1 - p^m) V^{max}$. This simple scheme assumes that patterns more common in the cultural corpus are associated with stronger and more precise event-based expectations. This approach resembles statistical learning schemes where predictions for rhythmic patterns heard frequently by a listener would be statistically robust [16], enabling accurate entrainment to (culturally) familiar rhythms [49] and exerting a stronger production bias during phase inference.

**Tractability.** While pPIPPET can be configured with an arbitrary number of expectation templates, it is computationally intensive to perform phase inference in the context of many templates at a fine temporal resolution (small time step $dt$). It seems unlikely that listeners would continuously update a probability distribution over all imaginable rhythmic patterns (templates), and instead might dynamically constrain this search space. Additionally, long-term learning might feasibly result in a more detailed (e.g. hierarchical) representation of event-based expectations than described here. Expanding the problem of pPIPPET to incorporate such cognitive tasks would be an obvious subject of future research.

For the practicality of simulations presented in this work, we constrain the number of expectation templates used by pPIPPET where appropriate. The procedure described above results in a set of 154 rhythmic patterns, each assigned a unique expectation template, and we run simulations at the resolution of milliseconds. For any three-interval stimulus rhythm, we constrain these patterns to the $N$ templates most similar to the stimulus (by Euclidean distance in $\mathbb{R}^3$). We set $N = 22$ for a balance between tractability and flexibility.

## Supporting information

**S1 Text. Simulation parameters.** Parameters used in all simulations presented, alongside other relevant configuration details.
(PDF)

**S2 Text. Derivation of filter equations.** Mathematical process for deriving the PIPPET and pPIPPET filters.
(PDF)

**S1 Fig. Tracking the phase of a 4:3 rhythm with different timing expectations.** Fig 4 extended to tracking the first five repetitions of the 4:3 rhythm.
(PDF)

**S2 Fig. Tracking the phase of a 1:1 rhythm with different timing expectations.** pPIPPET models configured as per Fig 4, tracking the first five repetitions of a 1:1 (isochronous) rhythm.
(PDF)

**S3 Fig. Tracking the phase of a 2:1 rhythm with different timing expectations.** pPIPPET models configured as per Fig 4, tracking the first five repetitions of a 2:1 (uneven) rhythm.
(PDF)

**S4 Fig. Expected patterns derived from the German and Turkish rhythm samples.** Prior likelihood of three-interval rhythms used to configure the pPIPPET filters in Experiment 2 and Experiment 3.
(PDF)

## Author Contributions

**Conceptualization:** Thomas Kaplan, Jonathan Cannon, Lorenzo Jamone, Marcus Pearce.

**Data curation:** Thomas Kaplan.

**Formal analysis:** Thomas Kaplan.

**Funding acquisition:** Thomas Kaplan, Lorenzo Jamone, Marcus Pearce.

**Investigation:** Thomas Kaplan.

**Methodology:** Thomas Kaplan, Jonathan Cannon, Lorenzo Jamone, Marcus Pearce.

**Project administration:** Jonathan Cannon, Lorenzo Jamone, Marcus Pearce.

**Resources:** Thomas Kaplan, Jonathan Cannon.

**Software:** Thomas Kaplan, Jonathan Cannon.

**Supervision:** Jonathan Cannon, Lorenzo Jamone, Marcus Pearce.

**Validation:** Thomas Kaplan, Jonathan Cannon, Lorenzo Jamone, Marcus Pearce.

**Visualization:** Thomas Kaplan.

**Writing – original draft:** Thomas Kaplan.

**Writing – review & editing:** Thomas Kaplan, Jonathan Cannon, Lorenzo Jamone, Marcus Pearce.

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
