## [Decision Letter · Decision Letter 0]

7 Mar 2022

Dear Mr Kaplan,

Thank you very much for submitting your manuscript "Modeling enculturated bias in entrainment to rhythmic patterns" for consideration at PLOS Computational Biology.

As with all papers reviewed by the journal, your manuscript was reviewed by members of the editorial board and by several independent reviewers. In light of the reviews (below this email), we would like to invite the resubmission of a significantly-revised version that takes into account the reviewers' comments. Please pay particular attention to the extensive and detailed comments of Reviewers 2 and 3.

We cannot make any decision about publication until we have seen the revised manuscript and your response to the reviewers' comments. Your revised manuscript is also likely to be sent to reviewers for further evaluation.

Sincerely,

Andrea E. Martin, Ph.D.

Associate Editor

PLOS Computational Biology

Samuel Gershman

Deputy Editor

PLOS Computational Biology

Reviewer's Responses to Questions

**Comments to the Authors:**

Reviewer #1: In their manuscript, “Modeling enculturated bias in entrainment to rhythmic Patterns”, the authors present a model of entrainment to time series of events: pPIPPET (an extension of PIPPET, from the same group) which is able to reproduce previously observed universal and enculturated biases in human entrainment. I consider the paper is well written and the model can help our understanding of how temporal expectations influence entrainment. I only have a few comments.

1.) I think the paper would benefit from reproducing the specific figures from the original studies reporting the results that the current model aims to replicate (if possible and with permission from the authors). This would facilitate the direct comparison and better assessment of the performance of the model.

2.) Fig 3B, can the authors report statistical results comparing phase uncertainty between the different models and ratios?

3.) Please check that PIPPET and pPIPPET acronyms are used as intended (e.g., line 346 and discussion). Sometimes the constant switch is a bit confusing.

4.) Throughout the paper, pPIPPET is presented mostly as a model of entrainment. However, the authors state in lines 181-183: “Note that we do not explicitly model behavioral entrainment: instead, we model how the listener would interpret the rhythm in light of their expectations, which we assume would inform their reproduction of the rhythm”. Could the authors expand more on this in the discussion? Is the intention to make a distinction between perceptual as opposed to motor entrainment? Could other factors mediating between the interpretation of the rhythms and its reproduction explain for example small differences between the model and empirical data in experiment 3?

Reviewer #2: Uploaded

Reviewer #3: Review of pPIPPET

The work is the extension of the PIPPET model to allow an additional layer of interference, the incorporation of multiple possible expected rhythmic patterns. The goal is to incorporate culturally-specific priors to model previously observed differences in rhythm categorizations. Overall a well written and very stimulating paper that provides a concrete model implementation that can model numerous recent results on cultural priors in the behavioural reproduction of simple rhythms. This model and these new extensions are exciting!

The following are general comments and then more specific comments. There were very few outright problems, and none were severe, so much of this can be read as suggestions to strengthen the impact of the paper. Many of the points raised were (I felt) belatedly addressed in the Discussion, so some of this would have been avoided by a better setup in the introduction.

The top issues for me were: treatment of the status of preferences for small integer ratios, ad-hoc choices regarding relationship between template priors and precision, and vagueness around perception vs. production.

==Modelling==

=Where does small-ratio preference come from?=

PIPPET has no small-integer bias built in and presumably would work equally well with rhythmic templates of arbitrary intervals. Here, the preference for small integer ratios is either assumed (Exp 1) or determined empirically by examining corpora. That is, it is _all_ down to experience but the phase estimation model itself has no constraints. This seems profoundly unphysiological. But maybe not--developmental work has showed that infants may have a 'pre-enculturated' phase in which they can discriminate metrical perturbations in small-integer-ratio patterns not prevalent in their culture. Cultures and sub-genres have rhythmic motifs that deviate essentially from simple integer ratios. However, the primacy of small-integer-ratio rhythmic systems would still seem to suggest there are some biological constraints within the brain that bias perception or production towards simple ratios of intervals. Perhaps it is an open question, but I feel it needs more discussion from the start.

Thus, there seems to be an intrinsic circularity, exemplified by the discussion at the top of p. 15, which acknowledges the brain has 'low-level' intrinsic priors, which are not actually modelled in the present work. It seems instead as if the approach is to take musical notation systems as a proxy for the brain's physiological priors and using music notation to externally impose what should rightly be an intrinsic prior of the model itself (if the model is intended to model cognitive operations). It seems backwards?

There is a nice discussion on p. 15 of extending to music as performed, though this does not solve this dilemma of where the quantal nature of rhythm lives--'out there' or 'in here'? There is further excellent discussion on p. 17 of the possibility that with practice, very subtle patterns can be learned, so this bug is actually a feature…but it still remains that simple ratios are easier to learn and perform, so the difficulty of creating templates deviating from simple ratios is something that ultimately needs to be accounted for.

=Predictions?=

The danger, then, as it is with any modelling work, is that the model just does what you tell it do to with the choice of priors and parameters. I don't think the work can be so simply dismissed but do feel that the value added of this modelling could be brought out more clearly from the beginning.

One way to do that is to discuss the small-ratio issue more completely. Coming to it only in the discussion makes reading up until then a bit more painful than it needs to be. Another way is to focus on other ways in which this model is fundamentally different from existing models, and thus what it adds. E.g., the very empirical results fit here have already been modelled using Bayesian ideas (e.g., in the early work by Honing, Desain and Sadakata). Where does this modelling effort fit in the space of existing models? There IS a nice discussion of this on p. 16, but it is hardly exhaustive of other modelling approaches. Maybe it's just a matter of preference, but I'd like these issues to be set up from the start. Yet another way to add value is to focus on some existing questions within the field that the model suggests answers to (and others may not). Another is to articulate clear predictions for future empirical studies. There were moves in these directions, but nothing really concrete or elaborated.

These are suggestions. I think this modelling effort to extend PIPPET is cool, novel (in the sense of explicitly modelling phase and its sophisticated formalization) and certainly worthwhile, but I feel all of these could be strengthened and would enhance the readers understanding and the appeal of the ideas here.

==The interrelation of priors and precision==

A different scheme for picking template prior probability and template precision is used in each experiment…Exp 1 uses different precisions and constant priors; Exp 2 uses fixed precisions and empirically-determined priors; Exp 3 uses inverse correlation between prior and precision. This seems ad-hoc and could stand a more unified presentation, or at least greater justification for the choices made in each simulation (or statements that the results do not essentially depend on the choices made). Some specific comments relating to this issue:

3.66 uncertainty is the _inverse_ of precision, no?

7.223: This is a critical point, that the 2:1 template was defined with lower precision. The reason for assigning lower precision should be justified more clearly--is it because such a template is less common and not as well 'learned' or something else? It has important implications for the evolution of the model's behaviour, e.g., a low-precision template is better able to attract unevenly timed events and thus drives the predictions of the model more strongly (all other things being equal)--this makes sense but is a little bit paradoxical on its face so it may be worth reminding the reader of the relationship between low precision and higher weighting of new evidence (which incidentally could have been better explained at 4.71).

As well as precision, the overall priors on the different templates must influence which one wins out. In the models for Experiment 1 the priors are equal, I believe. How is that motivated? It is natural to expect that a template with greater uncertainty for individual note timings should have a lower overall prior. While there is no such formal requirement mathematically, it has intuitive psychological merit and I think it would be a service to readers to articulate this topic more clearly. (Later, in Experiment 3 at 11.358 you do arrive at exactly this point, indeed linking higher priors and higher precision. So, if that is reasonable, why was it not used for all three experiments?)

The issue of priors comes up in passing also at 9.252. I'm not really sure why you suggest that model behaviour during the first few repetitions matters so much in determining the Malian 1:1 results of Figure 5B--after all, the 1:1 template quickly wins out.

How was the precision of the 4:3 template chosen? It looks simply intermediate between 1:1 and 2:1. Why? Also, for accuracy in exposition, it is worth pointing out that you (like Pollok) used 58:42 as the actual ratio, and explain why, as it is hardly a small integer ratio.

In experiment 2 priors are found empirically, but precision is held constant across templates. Why, exactly?

==Tempo-dependence==

The recent work by Polak et al found strong results of enculturation at a fast tempo. The current model is tempo-agnostic, so it would not be able to predict tempo-dependent effects such as found by Polak. This deserves comment. I see it is covered a bit in the discussion.

==Perception, Production, Both?==

It is unclear, throughout, whether the behaviour and model are meant to describe perceptual constraints, production constraints, or (as they may be fundamentally intertwined) both. In various places each is mentioned, but vaguely, so I didn't gain a coherent sense of the authors stance on this issue.

==Replication==

It is great service that the simulation code is made available through OSF.

==Contributions==

The manuscript would be enhanced by describing the contributions of each author.

==Specific points==

=Abstract=

"inferring a suitable learned pattern" could be clearer. As becomes clearer later, it seems like inferring _which_ previously culturally-learned pattern is most suitable to explain the current input.

=Introduction=

Overall, found the introduction a bit woolly--it gets across an idea, but the wording is rather imprecise.

P2.Line2: The first few sentences could stand reformulating. I think I get the gist of what you are trying to say, that people can abstract prototypical rhythms from variable temporal patterns of musical events (?) but there are too many concepts that are fuzzy: 'temporal stress patterns,' 'vary dramatically,' 'discrete rhythmic patterns'

2.6: unclear what 'which' refers to here--I think you intend it to mean that (what is) complexity depends on experience. Complexity doesn't seem like the right, or needed, term here, because it is culturally relative and can be defined in many ways--or do you have some more absolute pattern-based complexity metric in mind?

2.10: Throughout, would help to make clear if you are talking about the abilities of untrained or musically trained individuals (or dancers). I.e., implicit enculturation vs. trained.

2.29: An important point regarding interface between continuous rhythmic space and discrete space that can be (easily) written symbolically. Does this work intend to provide insights on the tendency of attraction to simple interval ratios, for example, which surely is cognitively prior to the ability to notate? This point was expanded upon above.

Surprised, however, to see the discussion at such a high level and not invoking early on the universal tendency towards simple ratios…Certainly the fact of attraction to integer ratios was known well before the Jacoby et al experiments cited in the next paragraph, albeit not studied across so many cultures. A more complete historical view of the literature would be appropriate. Refs 1 and 2 are part of the genesis of this entire approach and would seem more appropriate here than in the rather generic first sentences of the intro. Reading further, it's clear the authors are very familiar with the field, so making that more apparent in the intro would have avoided this criticism.

=Results=

It would be interesting to comment on how the models converged after the full 25 repetitions…i.e., did they look substantially the same as after the second repetition? The breadth of the distributions in Fig 5 suggest maybe not…that even in the 4:3 case the phases are being reset differently each repetition…[ah, there is noise added to stimulus times…not mentioned until the methods]

8.235: minor point but speaking about "phase corrections performed for all first intervals" is a little confusing--since phase is corrected by events and there is no explicit concept of interval in the model.

8.242: interesting that you are drawing a link between phase uncertainty in the model and tapping uncertainty in people. I really don't see the connection, since the model has a precise representation of mean phase. I suppose one might postulate that mu and sigma aren't represented explicitly in the brain at all, bur are merely parameters of convenience in the model (I.e., the phase could be represented across an ensemble of ramping/oscillating elements), so a tap would be drawn from some kind of distribution of phases that varies at each time point…?

9.258: Not sure what this means: "allowing pPIPPET to formally characterize behaviour that have previously been related to perceptual “prototypes” or “categories” for rhythm". There have been prior attempts to model using templates and maybe they should be discussed.

10.300: Curious about the justification for repeating each stimulus once. Experiment 1 showed that the selection of a template (in a simpler situation) took several stimulus repeats to begin to approach an asymptote in template posterior probability. Surely repeating each stimulus more than once is not very complicated.

10.304: Not totally clear what the model provides beyond the varying prior probabilities of different templates. Is the strength of attractor determined entirely by its prior, as suggested?

11.321: articulation of the model's ability to "generate specific hypotheses" needs more development.

11.344: in Experiment 3 using only one stimulus cycle to pick a template seems surprising and needs justification, e.g., was the winning template's probability significantly different from all others--i.e., was it always a clear choice? Did the iterated reproduction data show such rapid convergence to a stable pattern? Does this choice of method over-emphasize initially high-prior templates? If instead, waiting over a longer time span might a lower prior template emerge? Or is that never the case--Is the development of marginal probabilities for each template monotonic? It's hard to generalize from the examples shown in Figure 4, which involved at most 3 templates with equal priors. It would be very useful to see some analysis of the Experiment 3 simulations, not just the final results.

11.352: the phase 'at each event' in the previous stimulus _cycle_. Does this mean the phase estimate at the timestep prior to the events, or the updated phase at the timestep with the event?

13.378: clarify what is meant by "less stratified meter" here

16.528: Very interesting discussion. An issue with sensory updating models is that they tend to negate important timing deviations from the beat (establishing e.g., a laid-back feel) by assimilating them into posterior estimates. This might lead to optimal tracking of 'rhythm' but very much sub-optimal tracking of 'beat/meter' and feel. pPIPPET might offer a solution, to the extent that the agent is perceptually aware of the template.

14.422: "using an imprecise pattern of expectations" may be a poor word choice given the specific meaning of precision being part of another parameter group (3). Instead "using an inaccurate/non-matching template pattern" or something like that may be better.

17.559: ASAP is equally attributed to Patel and Iversen.

p. 17: The discussion of potential PFC involvement in templates is very well put together.

=Methods=

Overall, very clear. It would have aided the reader to know earlier about the various places in which noise was added, e.g., to the event times, as this was not clear in the body of the paper.

The determination of the meter templates from the probabilities of events at each subdivision of the measure is interesting as it pools across different scales (i.e., 111 could be three sixteenths in a row, three eighths, three quarters). Do you have a sense of how adding constraints might change the overall probability space? It would be very useful to see, in supplementary figures, ternary plots of the derived prior probabilities.

=Figures=

The figures are beautifully clear.

Figure 4: Why does uncertainty grow throughout the cycle in B, even at t=2 by which point the '4:3' model dominates?

Figure 5: Merely out of curiosity, how might you account for the slightly bimodal nature of the European distribution in A (i.e., the mini shelf on the left of the distribution).

Naturally it would be helpful to show the empirical data that this reproduces. However, consulting the paper shows a remarkable similarity that could be discussed more forcefully: not only do you find the large difference for 4:3 but you also capture the greater uncertainty at 2:1 than 1:1 (however, is this simply determined by your templates gaussian standard deviations, which vary in the same manner?). A rather secondary issue: The empirical data shows that the Malian uncertainty is greater for _both_ 1:1 as you reproduce but also 2:1, which you do not. Thoughts? Anyway, gets back to the issue of the interaction between priors and precision discussed above? It would be good to know how the specific parameter choices led to the fits, and what fell out of the model.

Figure 6: How might one account for the asymmetries in the classification results: e.g., the 112 catchment area is far more permissive of a range of intervals I and II than III (it is not a blob but is oriented northwest-southeast. Another: there is a distinct class for 233 but not 323. Is this (only) because each stimulus was presented once? What might it tell about the importance of order of intervals and how it interacts with the model? Do these details match experimental results?

Figure 7: It would be useful to explain more clearly why is KDE plotted here, vs entropy in Fig 6? I.e., how is this explained by the differences in task (perceptual categorization vs. iterated reproduction)?

Figure 8: Phase might better be labelled note onset position (in 16th notes) here, since elsewhere phase 1 = the end of the pattern.

**Have the authors made all data and (if applicable) computational code underlying the findings in their manuscript fully available?**

Reviewer #1: None

Reviewer #2: Yes

Reviewer #3: Yes

PLOS authors have the option to publish the peer review history of their article (what does this mean?). If published, this will include your full peer review and any attached files.

Reviewer #1: No

Reviewer #2: **Yes: **Edward Large

Reviewer #3: No
---

## [Decision Letter · Decision Letter 1]

21 Jul 2022

Dear Mr Kaplan,

Thank you very much for submitting your manuscript "Modeling enculturated bias in entrainment to rhythmic patterns" for consideration at PLOS Computational Biology.

As with all papers reviewed by the journal, your manuscript was reviewed by members of the editorial board and by several independent reviewers. In light of the reviews (below this email), we would like to invite the resubmission of a significantly-revised version that takes into account Reviewer 2's comments. Reviewers 1 and 3 are satisfied with the current version of the paper (note that Reviewer 3 requests a few clarifications).

We cannot make any decision about publication until we have seen the revised manuscript and your response to the reviewers' comments. Your revised manuscript is also likely to be sent to reviewers for further evaluation.

Sincerely,

Samuel J. Gershman

Deputy Editor

PLOS Computational Biology

Reviewer's Responses to Questions

**Comments to the Authors:**

Reviewer #1: The authors addressed all my comments. I am happy to recommend the paper for publication

Reviewer #2: Attachment uploaded

Reviewer #3: Thanks to the authors for the careful consideration of the review comments. It was an interesting read. I am overall satisfied with the changes. There were several points at which the authors considerately left the door open for future comment on my part and I do not feel so strongly about those issues as to request another round of review, or rather feel they are of interest to a rather small (but not zero!) set of readers, so I hope we'll have a chance to talk about it in the future.

Two clarifications: AR-R3.8.18: by 'sensory updating models' I was referring simply to the class of discrete models of phase and period updating based on event timing which are swayed entirely by external events. Yes, interesting discussion, particularly regarding rhythmic re-interpretation.

AR-R3.8.27 para 2: You are correct in your reading. However, I was 'looking' and comparing the _German_ and Malian group (sorry Bulgarians), perhaps biased by your later use of German folksongs to think of them as more representative of simpler meter folk. The statistical equality of Malians and Europeans at 2:1 is clearly driven by the Bulgarian respondents, while a Malian/German comparison would likely show differences at both ratios. It's a striking asymmetry present in your model between Europeans at 1:1 and 2:1, which is not seen in the (German) data. Sometimes insight can come from these details. Perhaps something worth pondering and including a sentence about if you feel you have enough information to make a meaningful comment.

**Have the authors made all data and (if applicable) computational code underlying the findings in their manuscript fully available?**

Reviewer #1: Yes

Reviewer #2: Yes

Reviewer #3: Yes

PLOS authors have the option to publish the peer review history of their article (what does this mean?). If published, this will include your full peer review and any attached files.

Reviewer #1: No

Reviewer #2: **Yes: **Edward W Large

Reviewer #3: **Yes: **John Iversen
---

## [Decision Letter · Decision Letter 2]

16 Sep 2022

Dear Mr Kaplan,

We are pleased to inform you that your manuscript 'Modeling enculturated bias in entrainment to rhythmic patterns' has been provisionally accepted for publication in PLOS Computational Biology.

Best regards,

Samuel J. Gershman

Section Editor

PLOS Computational Biology

Reviewer's Responses to Questions

**Comments to the Authors:**

Reviewer #2: The authors have addressed all my comments, and the paper is now ready for publication in my opinion.

**Have the authors made all data and (if applicable) computational code underlying the findings in their manuscript fully available?**

Reviewer #2: None

PLOS authors have the option to publish the peer review history of their article (what does this mean?). If published, this will include your full peer review and any attached files.

Reviewer #2: **Yes: **Edward W Large

---

## [Editor Report · Acceptance letter]

26 Sep 2022

PCOMPBIOL-D-21-02272R2 

Modeling enculturated bias in entrainment to rhythmic patterns

Dear Dr Kaplan,

I am pleased to inform you that your manuscript has been formally accepted for publication in PLOS Computational Biology. Your manuscript is now with our production department and you will be notified of the publication date in due course.

With kind regards,

Zsofia Freund
